# CLINB: A Climate Intelligence Benchmark for Foundational Models

## Abstract

Evaluating how Large Language Models (LLMs) handle complex, specialized knowledge remains a critical challenge. We address this through the lens of climate change by introducing CLINB, a benchmark that assesses models on open-ended, grounded, multimodal question answering tasks with clear requirements for knowledge quality and evidential support. CLINB relies on a dataset of real users' questions and evaluation rubrics curated by leading climate scientists. We implement and validate a model-based evaluation process and evaluate several frontier models. Our findings reveal a critical dichotomy. Frontier models demonstrate remarkable knowledge synthesis capabilities, often exhibiting PhD-level understanding and presentation quality. They outperform "hybrid" answers curated by domain experts assisted by weaker models. However, this performance is countered by failures in grounding. The quality of evidence varies, with substantial hallucination rates for references and images. We argue that bridging this gap between knowledge synthesis and verifiable attribution is essential for the deployment of AI in scientific workflows and that reliable, interpretable benchmarks like CLINB are needed to progress towards building trustworthy AI systems.

## 1 Introduction

A secure path towards Artificial General Intelligence (AGI) depends on the ability to effectively assess AI systems, to foster and track progress, as well as alignment with desired objectives (Russell, 2019). Significant research is dedicated to evaluation, and performance on popular benchmarks like Chatbot Arena (Chiang et al., 2024a) has become a primary driver of development. The rapid advancement of AI necessitates increasingly challenging benchmarks, which in turn demand technical sophistication and domain knowledge (Rein et al., 2024; Long, et al., 2025). Current 'hard benchmarks' are objectively difficult and beneficial to model improvement. However, the current approach to building these benchmarks has limitations. Firstly, tasks (prompts) often consist of esoteric puzzles and trivia, largely obscure to those outside a niche specialty. Secondly, to enable algorithmic verifiability, responses are typically limited to *closed form* tasks (Rein et al., 2024); e.g., short text, or numeric, answers and multiple-choice questions (Dinh et al., 2024; Justen, 2025; Long, et al., 2025). These represent only a fraction of real-world applications of Generative AI, and the challenges that arise from its use.

We focus on open ended, *generative*, question answering tasks and collaborative Human-AI environments. To understand, and improve, how foundational models handle climate change information we develop a benchmark, CLINB (**Cl**imate **In**telligence **B**enchmark), consisting of a dataset of real users' questions (Vaghefi et al., 2023) and a rubrics-based evaluation pipeline.[1] Climate change is a broad, complex and hotly debated topic, ultimately rooted in extensive scientific knowledge spanning multiple disciplines, from physical to social sciences. Moreover, the topic draws upon decades of institutional knowledge and best practices. CLINB's questions require research, evidence-assessment and synthesis skills. These fall under Long-Form Question Answering (LFQA) (Fan et al., 2019; Arora et al., 2025), where ensuring faithfulness and reliable attribution remain core challenges (Ji et al., 2023). For each question, multiple answers are curated by experts with advanced academic knowledge on the subject. Answers consist of free-form text, may contain visual content, and must be supported by robust evidence. All answers are evaluated by the experts in side-by-side experiments and the human feedback is used to compile grading rubrics which are validated

---

[1]We will release the benchmark data upon publication.

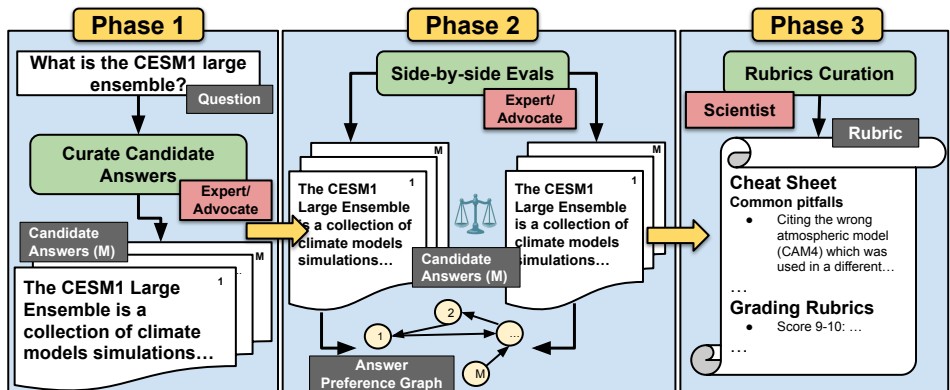

Figure 1: The multi-step, human-in-the-loop process to construct the CLINB dataset.

by climate scientists (among the co-authors of the paper) with experience as leads in main institutional reports.[2] The rubrics are then used for automatic assessment by a model-based *autorater*. This utilizes the "LLM-as-a-Judge" paradigm (Zheng et al., 2023; Arora et al., 2025), employing criteria-based approaches shown to improve alignment with human experts (Kim et al., 2024). The whole data creation process is performed in an AI-assisted tool.

We perform an empirical assessment of several frontier models. Our main contributions are:

**A New Expert-Grounded Benchmark for Scientific AI** We introduce CLINB, a benchmark for model-based evaluation of frontier models on complex, multimodal scientific communication. Its core is a new dataset of real-world climate questions paired with *data-driven, question-specific evaluation rubrics*, curated and validated by leading climate scientists through a novel three-phase, human-in-the-loop process.

**PhD-Level Synthesis vs. Attribution Failures** Frontier models demonstrate remarkable knowledge synthesis, often exhibiting a *PhD-level understanding*. However, this performance masks a critical inadequacy in grounding. We report substantial hallucination rates for references (10% to 25%) and even more failures for images (50% to 80% in certain settings), exposing a major gap between synthesis and verifiable attribution.

**Insights into Human-AI Collaboration Dynamics** Autonomous frontier models *surpass 'hybrid' answers* (curated by experts using weaker AI assistance), revealing the assisting model's capability—not human oversight—as the primary bottleneck. Counter-intuitively, highly motivated non-specialists (our 'Advocates') who deeply engage with AI tools can produce *higher-quality answers than domain experts* who engage less with AI during answer curation.

**A Validated Methodology for Scalable Oversight** We validate a rigorous, rubric-based autorater. Ablation studies demonstrate that structured prompts and *automated evidence-checking* are essential for mitigating inherent LLM judge biases. This process is hampered by inaccessible sources (up to 50%). Furthermore, we identify evaluation challenges, including *model familiarity bias* in human raters and the limitations of *rubrics to generalize* across models.

## 2 CLINB DATA

Here we explain the data creation process underlying the CLINB benchmark. The data consist of human-curated questions, answers, pairwise preferences over pairs of answers and finally question-specific answer-grading instructions: the rubrics. The data is curated by three groups of human experts: Advocates, Experts and Scientists, in three phases illustrated in Figure 1.

### 2.1 HUMANS IN THE LOOP

Human expertise is critical for assessing AI in knowledge intensive tasks, and is a scarce resource(Rein et al., 2024; Long, et al., 2025). We intentionally organized a group among the authors,

---

[2]E.g., IPCC (https://www.ipcc.ch/) and NCA (https://toolkit.climate.gov/NCA5).

which we call the *Scientists*, to validate the study. This group consists of six academics. Five are climate scientists with extensive expertise and multiple lead roles in IPCC and NCA reports, and the sixth is a leading expert in Climate Finance.

The Scientists curated parts of the data and oversaw the scientific validity of the work, but do not have the capacity to produce sufficient amounts of data. To scale human data collection we formed a pool of 40 experts by directly recruiting active academics (mostly PhD students and postdocs) with the necessary domain expertise. We call this group the *Experts*. To increase diversity and exploration we further recruited 17 raters from the Climate Fresk community, an NGO organizing workshops to explain IPCC reports to the public.[3]. We call this group the *Advocates* (cf. Appendix B.1).

## 2.2 QUESTIONS

CLINB's questions are sampled from the logs of `chatclimate.ai` (Vaghefi et al., 2023) a chatbot dedicated to climate change which has collected thousands of users' questions. These questions are challenging because they involve technical topics with complex interdisciplinary ramifications. Compared to trivia and puzzles, real users' questions express genuine information needs. At the same time questions can be poorly worded or unclear, making it difficult to even interpret the question or the context in which it is posed. Furthermore, the role of evidence is critical. The ability to properly assess answers to such questions may be directly useful for advancing AI-assisted science, science communication and decision-making in the real world.

To match our Scientist group's expertise, we selected questions from six key topics: 'Weather and Climate Extremes', 'Mitigation Pathways', 'Detection, Attribution and Uncertainty', 'Climate Finance and Risks', 'Climate Impacts, Adaptation and Vulnerability', and 'Climate Change Scenarios'. For each topic, Scientists chose approximately 30 questions and assigned one of three difficulty labels: High Confidence (answerable from authoritative sources), Advanced (requiring technical sophistication), or Open (actively debated). We also classified each question by its relevant IPCC working group (WGI: Science, WGII: Adaptation, WGIII: Mitigation).[4] The final dataset represents all topics and working groups, with a slight prevalence of climate science questions (WGI) and the 'Impacts', 'Detection', and 'Scenarios' topics. High Confidence (35.7%) and Advanced (35.1%) questions are the most frequent, though the fraction of Open questions is substantial (29.2%). These properties are summarized in Figure 4 (Appendix).

### 2.2.1 ANSWER FORMAT

Answers include three components: *text*, *images* (optional), and *references*. Typically, 500 words are sufficient for an initial, yet substantial, answer and we use this as the length limit for the main body of the response.[5] All key points in the answer that quote, explicitly or implicitly, external sources must be accompanied by citations, with the references listed in a dedicated section. We encourage the inclusion of images, e.g. to summarize quantitative information.

## 2.3 DATA CONSTRUCTION PROCESS

As in recent work (Ruan et al., 2025; Arora et al., 2025), we design a rubric-based assessment task. Rubrics provide a level of abstraction that can contribute to scalable, accurate and consistent, model-based evaluation, moving beyond "vibe-based" assessments (Robertson & Koyejo, 2025). We want the rubrics to be data-driven, specific to technical points, common pitfalls and misconceptions. To this end, we implement a data processing pipeline, see Figure 1, which emphasizes a transparent and traceable curation and assessment process.

**Phase 1 - Candidate answers:** For each question, a human produces a candidate answer in two steps: (i) by curating an *outline* (ii) by curating a full *first draft* answer derived from the outline. The first versions, of both outline and full draft, are generated by the model following an iterative self-improving, retrieval-augmented, process. We produce $N \geq 3$ hybrid answers for each question. 92% of the candidates are curated by Experts, 8% by Advocates. We add to the set an *LLM answer*, produced independently by the model, and a *merged* answer. This is produced by the model by synthesizing the existing candidates. The total is $N+2$ answers. Overall, we collect 1330 candidate

---

[3]`https://climatefresk.org/world/`.

[4]Cf., `https://www.ipcc.ch/working-groups/`.

[5]Answers should be concise and to the point, also to lighten the cognitive load on human raters. A 500 words answer can typically be displayed in full on most screens without scrolling.

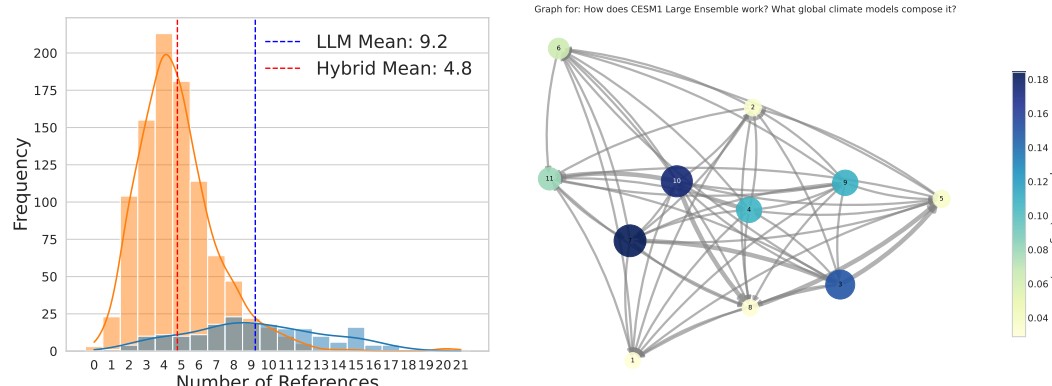

(a) References counts in candidate answers.  (b) Graph with weighted edges (counts) and ranked nodes.

answers. 82.6% have images. 99.7% have references – 6.1 on average; LLM answers have roughly twice as many references as hybrid answers (Figure 2a). See also Appendix A.2 for more details.

**Phase 2 - Pairwise Answer Preferences**   To learn the different candidate answers' strengths and weaknesses, we run side-by-side evaluations where humans evaluate which answer in a pair is better and why, a standard approach for collecting human preference data (Chiang et al., 2024b). We provide a form, derived from (Bulian et al., 2024) where the graders can select aspects where either answer is superior. At least one dimension must be selected on the preferred answer side. Humans do not evaluate answers they have curated. We collect 8654 preferences, 2.17 on average per unique answer pair. 70% of the pairwise preferences are from the Experts and 30% from the Advocates.

Due to the scarcity of human raters, it is not possible to collect numerous preferences on answer pairs. In addition, due to the complexity and nuances of the task, we do not expect that this process would converge to the identification of clear-cut good and bad answers. Instead, we deal with the topic of the question holistically, relying on the full set of answers and the intrinsically multi-dimensional human preference feedback. We represent the pairwise preferences as a directed graph with edges weighted by the preference counts. This provides tools for computing characteristics of the data that can be used to gain insights for compiling the grading rubrics. Figure 2b plots an example showing how the graph structure can be complex and densely connected. Most graphs form one large strongly connected component, indicating the presence of intransitivity or low confidence regions. However, one can still find informative structure, such as sinks and sources (worse and better answers) and groups of high/low ranking answers.

**Phase 3 - Question Rubrics**   In the last phase of the process, each question is associated with a *rubric*, to be used for assessing answers in a transparent and consistent way. Our rubrics consist of a 'Cheat Sheet': a compact reference for what to look for in the answers, and a 'Grading Rubric', explaining the grade bands. We find the use of explicit grading useful, even if for practical reasons we mostly rely on pairwise evaluations, because it forces the explanation of decisions in a more interpretable and calibrated way. An initial draft of the rubric is generated by Gemini 2.5 Pro, using a prompt which lists all the answers, including images and references, the quality guidelines, the pairwise preference graph and other instructions.

Figure 2b shows the preference graph for the question "How does CESM1 Large Ensemble work? What global climate models compose it?". The graph forms one strongly connected component, indicating disagreements, mostly deriving from different prioritization of criteria among human raters. However, the model is capable of surfacing critical feedback from raters. For example, the model identifies the possible confusion between correct and incorrect atmospheric model components. To break ties and propagate non-local preferences, we rank answers/nodes. The model is able to propose an holistic synthesis of the data in the generated rubrics, including using the images content (cf. the example in Appendix A.4).

The last step involves the Scientists, who manually curate the final form of the rubrics. In general, the AI-generated rubrics provide an impressive starting point. Areas for improvement concern greater scientific nuance and rigor: sometimes the rubrics lacked depth or overlooked critical details – e.g, including overshoots scenarios for a question on limiting warming to below a certain target. Initial

rubrics can also reflect an overconfidence bias present in the published literature, where positive findings are over-represented. Another aspect to improve upon is the critical assessment of the question's premises, whether the question makes sense, is well-formulated and clear.

## 2.4 AI-Assisted Curation Tool

All steps above are performed in a dedicated AI-assisted user interface, called the *Editor*. The purpose is to both facilitate and analyze the 'human-in-the-loop' process. The tool allows experts to carry out all the three phases in dedicated workflows supported by a context-aware LLM assistant. Additionally, the Editor provides direct access to web search for both documents and images, as well as ranking and understanding (e.g., summarization) of the retrieved content. For publications listed in OpenAlex[6], we include bibliographic metadata. The tool accepts feedback in both structured and natural language. We use Gemini 2.5 Flash as the assistant AI and Google for web search.

## 3 Experiments and Analysis

We evaluate several publicly available models: OpenAI's GPT-5 and o3, Google's Gemini 2.5 Pro and Flash, and Anthropic's Claude Opus 4.1 and Sonnet 4. Models are accessed via their public APIs with default settings, without search enabled. For each system, we submit one request per question in the dataset using a 'system prompt'. We also evaluate the highest ranking hybrid answer from our dataset. The 'system prompt' provides the question, information about the expected answer format, the quality dimensions, the role of references, visual content etc. The prompt specifies clearly that each piece of evidence must be accompanied by a URL (cf. the full prompt text in Appendix D.2).

### 3.1 Model-Based Pairwise Assessment

We perform model assessment of answer pairs: two answers are evaluated side by side (SxS) to identify the preferred one. We adapt the Chatbot Arena's battles setup (Chiang et al., 2024a). A single battle involves one question and two systems answers. We run three rounds of evaluations. In each round, we run two pairwise evaluations, swapping the order of the answers to control for position bias (Wang et al., 2023). The system with the majority score is the winner, otherwise it is a tie. This procedure defines a single battle. We compute battles between all pairs of sources answers, 4147 battles total. We estimate the ELO scores for all systems using the Bradley-Terry model, including a bootstrap $95\%$ confidence interval.

As the judge model we use Gemini 2.5 Pro with a temperature of $0.7$. The model relies on an 'assessment prompt' which includes the general quality guidelines, the question-specific rubric, the input data (question, answer pair) and the task instructions. The latter include a required explanation for the decision aligned with the quality dimensions of Section 2.3. Images are encoded in place, in the answer, as bytes, if the image content can be fetched via the link. To assess the evidence we check the validity of the images and references links in the responses. We classify the returned status as either a valid web page, whose content may or may not be accessible; e.g., due to paywalled content, or as invalid, due to the URL being hallucinated. We instruct the model to factor in this information while assessing the answers' statements. See Appendix D.3 for the full prompt. We refer to the judge model as the **CLINB autorater**.

### 3.2 Experiments Findings

Table 1 summarizes the experimental results.[7] We report ELO scores for overall pairwise preference (Answer SxS) and per quality category dimension (Answer Dimension). The upper table contains the results for the CLINB autorater. The lower table reports the results from a manual validation experiment for a sample of 1976 battles from the top 5 systems outputs, using the protocol of Phase 2 (Section 2.3). The manual assessment, 'Answer SxS/Experts' and 'Answer Dimensions (Experts)' in Table 1, relies only on the Experts group, three raters per battle. Since the autorater and the Experts reveal systematic disagreements we sampled a number of disagreement battles for deeper analysis. The Scientists repeated the pairwise human evaluation for 72 of these battles, see 'Answer SxS/Scientists' in Table 1.[8]

---

[6] https://openalex.org/.

[7] In Appendix F, we discuss specific examples of question-answer pairs and their evaluation.

[8] Appendix C.4 and Appendix C.3 report results at more granular quality dimensions and by question types, which broadly align with the general findings reported here.

| | CLINB Autorater | | | | |
| | Answer SxS | Answer Dimensions | | | |
| System | | Citations | Images | Knowledge | Presentation |
| GPT-5 | $1150 \pm 19$ | $1104 \pm 9$ | $905 \pm 33$ | $1167 \pm 8$ | $1106 \pm 15$ |
| Claude Opus 4.1 | $1135 \pm 19$ | $1219 \pm 10$ | $965 \pm 22$ | $1153 \pm 7$ | $954 \pm 13$ |
| GPT o3 | $1018 \pm 18$ | $846 \pm 7$ | $785 \pm 33$ | $1066 \pm 7$ | $1349 \pm 23$ |
| Gemini 2.5 Pro | $969 \pm 18$ | $949 \pm 8$ | $970 \pm 19$ | $954 \pm 6$ | $960 \pm 20$ |
| Hybrid | $945 \pm 18$ | $913 \pm 7$ | $1358 \pm 21$ | $868 \pm 6$ | $749 \pm 16$ |
| Claude Sonnet 4 | $915 \pm 19$ | $981 \pm 7$ | $822 \pm 30$ | $885 \pm 6$ | $861 \pm 19$ |
| Gemini 2.5 Flash | $868 \pm 18$ | $875 \pm 8$ | $798 \pm 30$ | $813 \pm 7$ | $803 \pm 15$ |
| | Human Assessment | | | | |
| | Answer SxS | | Answer Dimensions (Experts) | | |
| System | Experts | Scientists | Citations | Images | Knowledge | Presentation |
| GPT-5 | $906 \pm 18$ | $1040 \pm 115$ | $970 \pm 8$ | $633 \pm 31$ | $975 \pm 8$ | $902 \pm 8$ |
| Claude Opus 4.1 | $1115 \pm 20$ | $1149 \pm 111$ | $1244 \pm 12$ | $862 \pm 22$ | $1067 \pm 8$ | $1098 \pm 9$ |
| GPT o3 | $950 \pm 20$ | $959 \pm 114$ | $808 \pm 9$ | $716 \pm 27$ | $1078 \pm 8$ | $935 \pm 7$ |
| Gemini 2.5 Pro | $1015 \pm 18$ | $943 \pm 144$ | $1043 \pm 9$ | $1042 \pm 20$ | $970 \pm 7$ | $1062 \pm 8$ |
| Hybrid | $1015 \pm 20$ | $910 \pm 115$ | $842 \pm 8$ | $1486 \pm 25$ | $831 \pm 9$ | $922 \pm 8$ |

Table 1: ELO scores and 95% confidence intervals for the CLINB Autorater and human assessments. The models in the lower table are sorted in the same order as in the upper table.

**Humans/Autorater Agreement** The main disagreements between Experts and the CLINB Autorater concern the quality of the Hybrid and Gemini 2.5 Pro answers, on one side, and that of GPT-5, on the other. On close inspection, we find the Scientists to agree with the CLINB Autorater. In particular, they prefer GPT-5 and Claude Opus 4.1 for 'Knowledge' and 'Presentation'. We are inclined to attribute the Experts preference to familiarity bias, due to overexposure to Gemini and hybrid outputs in Phase 1 and 2 of the data creation. The Experts may have become accustomed to the presentation style of Gemini, while that of GPT-5 is more terse and relies more heavily on mathematical notation. GPT-5, and also OpenAI o3, are also penalized by the Experts for the lack of images, a strength of the Hybrid and Gemini 2.5 Pro answer, which is however a secondary epistemological factor for the Scientists.

**Hybrid Answers** Hybrid answers are better than those of the underlying LLM, Gemini 2.5 Flash – which is consistent with results from Phase 2 (Appendix A.2). They are also better than Claude Sonnet 4's answers, and not far behind Gemini 2.5 Pro. However, Hybrid answers are ranked lower than top models, by both autorater and Scientists. They are worse in terms of knowledge and presentation by autorater, Scientists and Experts. They rank lower also in presentation and citations quality. This indicates that the quality of the model in the human-in-the-loop framework is crucial, and better models exceed the quality of hybrid methods based on lower-performing models.

We also note that the Advocates group, who participated only in Phase 1 and 2, produced high-quality answers. Experts rated the Advocates' answers higher than any other answer source, including their own (cf. Appendix B.1.1). Evidence from the Editor use shows that Advocates engaged much more with all aspects of curation, including AI assistance. This suggests that high motivation and AI assistance can be an effective combination.

**Frontier Models' PhD-Level Knowledge** Based on the experiments, and close qualitative inspection, it is clear that the frontier models' knowledge quality is remarkably high. On the CLINB data, in terms of knowledge and presentation, Claude Opus 4.1, the two OpenAI models and Gemini 2.5 Pro, match or exceed the performance of PhD level humans.

**Ablations** We perform several ablation studies with the autorater (Table 4). Notably, removing the question-specific rubrics from the prompt changes the results only in the bottom half, with the Hybrid answers overtaken by Gemini 2.5 Flash and Claude Sonnet 4. This suggests that the additional resolution provided by the rubrics applies primarily to the kind of responses used to develop the rubrics. Or, in other words, that rubrics are far from complete. Hence, it is important that rubrics adapt to new data as better models become available. While the system-level scores do not change

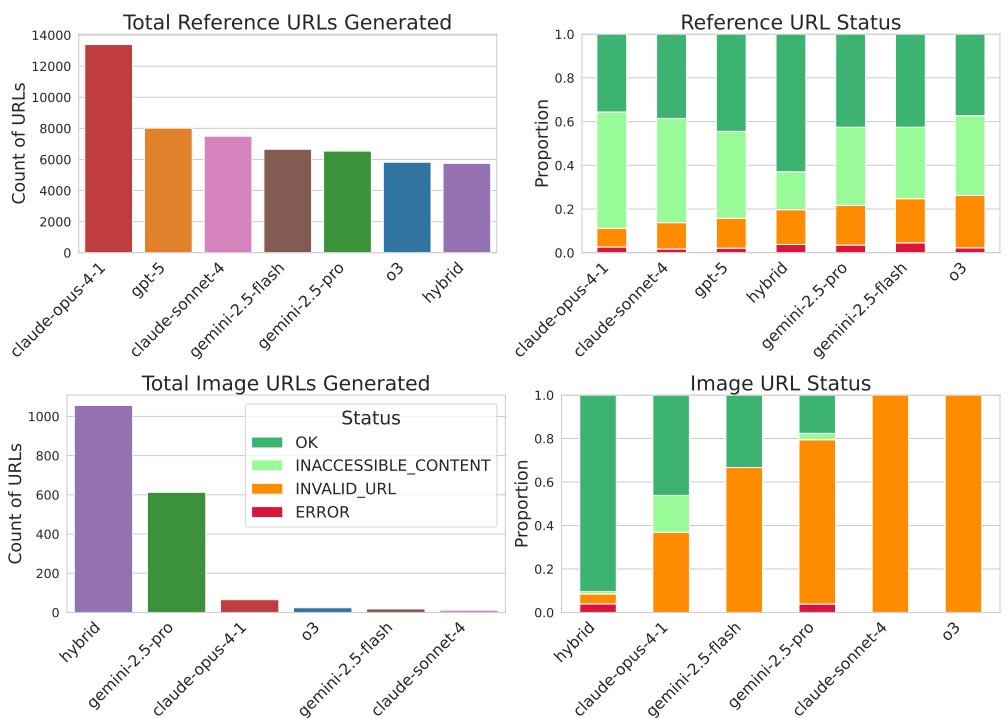

Figure 3: Number of reference (top), and image (bottom), URLs and their status.

much without rubrics, we notice a trend in the justifications of individual ratings. The autorater with rubric often assigns high scores to *ungrounded* information that correctly matches the rubric requirements, while the rubric-free autorater follows the general requirement to treat any claims that are not supported by references as non-existent more strictly. Thus, removing the rubrics slightly shifts the assessment focus from knowledge quality towards mechanistic verifiability.

Other ablations show that the autorater moves the judge model away from intrinsic biases, including favoring style over substance Gudibande et al. (2024) and AI-AI bias (Panickssery et al., 2024). The evidence instructions and the quality of the judge model have also a large influence.

**Evidence Quality** Figure 3 summarizes numbers and status of evidence links. Claude Opus 4.1 is by far the best for quantity and validity of citations. It has also the highest rate of pay-walled content, thus likely peer-reviewed sources. It is followed by GPT-5, Claude Sonnet 4 and Gemini 2.5 Pro. Roughly $25\%$ of OpenAI o3's URLS are hallucinated, corroborating recent findings on the persistence of citation errors in LLMs (Byun et al., 2024; Tang et al., 2024). Experts and the CLINB Autorater give low scores to Hybrid answers (which have the fewest citations), highlighting the gap between AI and humans in terms of literature processing capacity.

For images the results are even clearer. Hybrid answers stand out for superior visuals, while LLMs' answers often have very poor choice of visuals, or none. Multimodal understanding and composition shows considerable headroom for models. Among models, Gemini 2.5 Pro has the best performance on image quality according to both Experts and autorater. It generates a number of hallucinated URLs but also far more links than the next system, OpenAI o3. The CLINB autorater image scores for GPT-5 seem somewhat unjustified, given that it does not provide image links and may point to *anchoring bias*: the judge model has a hard time penalizing an overall strong output. If the inclusion of images is made mandatory, the hallucinated links rates range between $50\%$ and $80\%$ (Figure 9).

**Informal assessment of strengths and weaknesses** In addition to pair-wise preferences the autorater also produces a detailed assessment across question specific and shared rubric dimensions which can be used for further analysis. As the amount of text is too large for manual analysis, we use Gemini 2.5 Pro to extract recurring patterns and examples from the justification. Hence the characterizations in Table 2 and Appendix E should be considered *qualitative and informal*, however they are in line with the results of Table 1 and Appendix C.4.

Table 2: Informal strengths (+) and weaknesses (-). See also Appendix E.

|  | GPT-5 | OpenAI o3 | Claude Opus 4.1 | Gemini 2.5 Pro | Hybrid |
|---|---|---|---|---|---|
| Question Interpretation |  |  | - | + | + |
| Facts and Numbers | + | + | + | - | - |
| Depth & Nuance | + | + | + | - | - |
| Grounding & Images |  | - | + | + | + |

## 4 RELATED WORK

As foundation models achieve high performance on general knowledge tests like MMLU (Hendrycks et al., 2020), the research community has pivoted to more challenging evaluations to differentiate model capabilities. This includes benchmarks testing advanced reasoning with graduate-level or "Google-proof" questions, such as GPQA (Rein et al., 2024), HLE (Long, et al., 2025), and OlympiadBench (He et al., 2024). Concurrently, there is surging interest in "AI for Science" (Eger et al., 2025), leading to the development of specialized "Science LLMs" (Zhang et al., 2024) and benchmarks across diverse domains. This includes efforts in medicine (Arora et al., 2025), biology (Laurent et al., 2024; Justen, 2025), chemistry (Bran et al., 2023; Malikussaid & Nuha, 2025), and the development of AI research assistants or "AI Scientists" (Xie et al., 2025; Liu et al., 2025). This trend extends to specific scientific skills, such as multimodal understanding and figure interpretation (MMMU (Yue et al., 2024), SciFIBench (Roberts et al., 2024)), scientific coding (SciCode) (Tian et al., 2024), and tool-augmented scientific reasoning (SciAgent) (Ma et al., 2024). While these benchmarks establish a high level of difficulty, often claiming expert-level performance, they frequently rely on multiple-choice or short-answer formats for algorithmic verifiability (Justen, 2025; Dinh et al., 2024). In the domain of climate change, there is work on AI alignment (Kaack et al., 2022; Rolnick et al., 2022), and efforts like ClimaQA (Manivannan et al., 2024) and AtmosSci-Bench (Li et al., 2025) that use expert-level content but are similarly constrained by traditional formats. CLINB fills a critical gap by focusing on the complexity of synthesizing and communicating established scientific knowledge (Bajpai et al., 2024) in response to real user queries (Vaghefi et al., 2023).

A central challenge for generative models is producing extended, coherent, and verifiable text. Early Long-Form Question Answering (LFQA) datasets like ELI5 (Fan et al., 2019) established this task, often addressed using Retrieval-Augmented Generation (RAG) frameworks (Lewis et al., 2020). This evolved into models explicitly designed to generate answers supported by retrieved evidence and citations, such as WebGPT (Nakano et al., 2021) and GopherCite (Menick et al., 2022). However, ensuring faithfulness and avoiding hallucination remain core problems (Ji et al., 2023). The need for reliable grounding has led to a proliferation of research focused on citation and attribution. Recent studies highlight persistent issues, including the generation of non-existent references (Byun et al., 2024) and poor citation quality, particularly in long-context scenarios (Tang et al., 2024; Zhang et al., 2025). This has motivated new benchmarks, including general-purpose evaluations like ALCE (Gao et al., 2023) and ASQA (Stelmakh et al., 2022), as well as specialized frameworks focusing on long-context QA (LongCite (Zhang et al., 2025), L-CiteEval (Tang et al., 2024)) and medicine (MedCite (Wang et al., 2025), (Wu et al., 2025)). Various methods aim to improve attributed generation, such as Chain-of-Thought prompting (Ji et al., 2024), preference learning (Li et al., 2024a), self-reflection and critique (Asai et al., 2024), and transparent utilization of internal and external knowledge (Shen et al., 2025). Beyond autonomous generation, CLINB also investigates answers created through Human-AI collaboration, a critical area of study as AI integrates into knowledge work (Treude & Gerosa, 2025). Research shows that while AI assistance can improve productivity, the quality of the base model and the nature of the collaboration significantly impact the final output (Noy & Zhang, 2023). Furthermore, the methodology of using experts-in-the-loop to create the benchmark aligns with dynamic benchmarking strategies designed to keep pace with model advancements (Kiela et al., 2021). While benchmarks like ExpertLongBench (Ruan et al., 2025) and HealthBench (Arora et al., 2025) also use rubric-based evaluation, CLINB is unique in its focus on a combination of synthesis, attribution, multimodality, and hybrid generation.

Evaluating complex outputs at scale has motivated the "LLM-as-a-Judge" paradigm as a scalable complement to human evaluation (Zeng et al., 2023; Li et al., 2024b; Gu et al., 2024). Strong models can achieve high alignment with human preferences in pairwise comparisons (Zheng et al.,

2023), leading to methodologies like comparative assessments with ELO rating systems, popularized by Chatbot Arena (Chiang et al., 2024a). However, the reliability of this paradigm is a subject of intense research, with efforts dedicated to benchmarking the judges themselves for ranking capabilities (Gera et al., 2025). Studies have identified critical inherent biases, including position bias (favoring the first-presented answer), verbosity bias, and self-enhancement bias (Wang et al., 2023; 2024). Furthermore, recent work has shown a systematic "AI-AI bias," where models prefer LLM-generated text over human-authored text, regardless of objective quality (Laurito et al., 2025; Panickssery et al., 2024), and that pairwise comparisons can amplify these biased preferences (Jeong et al., 2024). Other work cautions that high alignment may be misleading, as judges might favor stylistic similarity over substantive correctness (Gudibande et al., 2024), that critiques generated by judges can be flawed (Sun et al., 2024), and that a model's ability on a task does not guarantee its ability to evaluate it (Oh et al., 2024). This growing body of evidence highlights the fragility of naive LLM-as-a-Judge implementations, and the need for rigorous safeguards.

In response to these challenges, our work aligns with a research thrust focused on increasing the methodological rigor of LLM-based evaluation (Calderon et al., 2025). The core of CLINB's contribution is a transparent, expert-in-the-loop process for generating fine-grained, per-question rubrics, moving beyond "vibe-based" assessments (Robertson & Koyejo, 2025). This approach can be viewed as a form of scalable oversight, leveraging model assistance to apply expert-defined criteria consistently (Bowman et al., 2022). This criteria-based approach is validated by several studies showing improved alignment with human experts. Kim et al. (2024) (Prometheus) trained a specialized evaluator LM on custom rubrics. Other approaches focus on decomposing evaluation into atomic facts for fine-grained assessment (e.g., FActScore) (Min et al., 2023). Other frameworks leverage checklists (RocketEval (Wei et al., 2025)) or combine rubric-based assessments (Hashemi et al., 2024) with reinforcement learning to mitigate biases, particularly in scientific domains (YESciEval (D'Souza et al., 2025)). Fine-tuning judges can also yield scalable and accurate evaluators (e.g., JudgeLM) (Zhu et al., 2025). The reliability of the judges themselves remains crucial, emphasized by benchmarks like JudgeBench (Tan et al., 2025). CLINB's methodology directly addresses the known biases of LLM judges by anchoring our evaluation against the highest level accessible of human domain expertise.

## 5 CONCLUSION

Human and AI-based assessments suggest that frontier models have reached PhD-level performance in knowledge and presentation on advanced climate change topics. However, this competence is undermined by hallucinations of images and references. To improve traceability and trust, models must provide better explicit evidential support for their responses. Our next steps include evaluating search-enabled systems, making rubrics adaptive, and applying deeper scrutiny to evidence. As we envision the best future models to incorporate advanced skills, including dataset retrieval, statistical analysis, and image synthesis, the benchmarks have to co-evolve.

The evaluation of generative responses in deep-expertise regimes is a difficult, often ill-defined (Who is the audience? What was the original intent?), expensive, and fundamentally unscalable task for humans. We propose that collaborative frameworks, where experts and AI perform meaningful end-to-end tasks together, are key to scalable performance tracking. Climate change is a particularly well-suited domain for this approach, as its institutional practices are built on collaboration, consensus-building, and the rigorous assessment of evidence.

Our findings, however, reveal a critical challenge: while human-in-the-loop curation improves weaker models, frontier models can already surpass these hybrid results autonomously. A central challenge is the design of interfaces that enable expert-AI collaboration, perhaps through continuous interaction and mutual questioning, to achieve a synergistic performance that exceeds what the model can do alone. Such a framework inherently cultivates essential human skills: the critical assessment of AI-generated content, the verification of claims against ground-truth data, and the identification of model biases. Ultimately, this new mode of collaboration trains humans to be more sophisticated thinkers, strengthening our collective ability to validate information and build robust knowledge in an age of AI.

## 6 REPRODUCIBILITY STATEMENT

Upon publication we plan to release the benchmark data. We provide the prompts used in the experiments in Appendix D, and the details of the model assessment process and the model used as judge in Section 3.1.

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

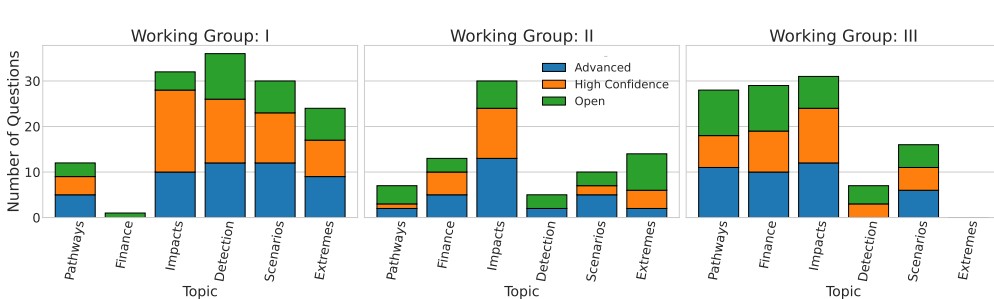

Figure 4: Distribution of topics and difficulty level by working group.

# APPENDIX

## A DATA

### A.1 QUESTIONS

The initial dataset consists of real-world user questions collected from `www.ChatClimate.ai`, a platform that provides climate-relevant information grounded in the IPCC Sixth Assessment Report (AR6). The questions represent authentic user inquiries submitted through the platform's conversational interface, which employs a Retrieval-Augmented Generation approach (Vaghefi et al., 2023). The data is publicly accessible on `https://doi.org/10.5281/zenodo.13788803`.

Figure 4 summarizes the main dimensions of the final CLINB questions set: IPCC working group, difficulty level and topic.[9]

### A.2 CANDIDATE ANSWERS

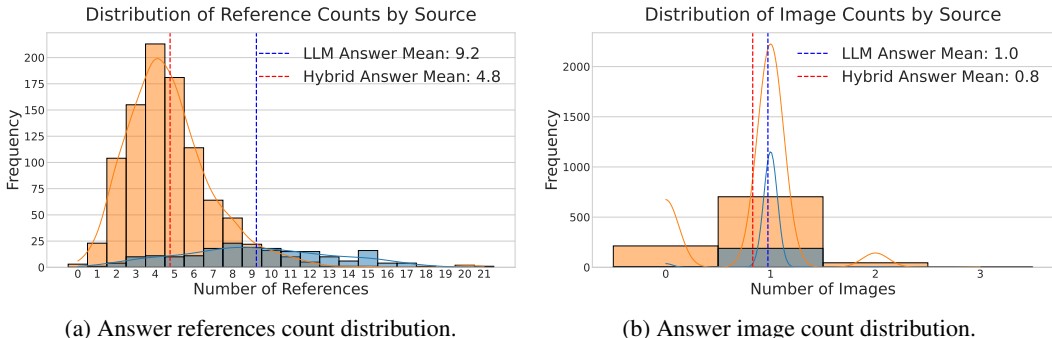

(a) Answer references count distribution.  (b) Answer image count distribution.

Figure 5: Summary of benchmark results for various leading models.

Figure 5 reports statistics relative to the number of references and images in the candidate answers data. Overall, there are 8086 references, the mean number of references for Hybrid answers is 4.8 and 9.2 for the 'LLM' answers, cf. Figure 5a. Hence, human experts tend to cut down substantially the references suggested by the LLM. The total number of images in the candidate answers set is 1176, the mean number of images in 'LLM' answers is 1.0 and 0.8 for the 'Hybrid' answers. Figure 5b plots the frequency distributions. Also for images, humans remove images when they are not considered necessary, or a good image cannot be found.

### A.2.1 LLM ANSWER GENERATION

**Answer Outline** The initial answer outline is created by prompting the model to generate a structured plan, limited to max 5 points, for writing a short essay on the topic of the question.

---

[9]Each question can be assigned multiple labels, for each category.

Figure 6: Distribution of best answer source, Hybrid or LLM, and evidence: images, references.

**First Draft Answer**   A full answer is generated by prompting the model to transform the user-curated outline into a full-prose answer of approximately 300 words. The context regarding the interaction with the user while curating the outline, is included in the prompt.

**LLM Answer**   The LLM answer added to the set of candidate answers is created as follows:

1. An initial answer is generated using only the user's question as the prompt;

2. We prompt the LLM to suggest search queries about the main keypoints in the answer;

3. The queries are issued to Google, to obtain both document and image search results.

4. The LLM filters out non relevant or low quality results and ranks the rest.

5. The LLM improves the answer using, and citing as needed, documents and images.

**Merged Answer**   The 'Merged' answer is generated using the LLM, as follows:

1. In the prompt we list the question and all the answers, including images and references;

2. We ask the model to merge all the keypoints made in the answers in a coherent way.

   • The model should resolve conflicting keypoints by consulting the supporting evidence.
   • In the absence of a clear resolution, different views should be included and explained.

3. We ask the model to review, reconcile and renumber the references appropriately, so to keep only those required by the answer. Similarly for the image(s).

A.3   PAIRWISE ANSWER PREFERENCES

We evaluate the candidate answers quality from the preference graphs. The answer graph encodes preferences as directed edges between answer pairs, weighted by the counts on that preference. To rank the answers, we compute the pagerank score (Page et al., 1998) for each node. As ranking metrics, we report Mean Reciprocal Rank (MRR) and Precision@1. The table below shows that the Hybrid answer is the best 70.3% of the time, the LLM answer is preferred 19.1% of the time. The Merged answer is the worst (10.5%).

| Answer Source | Precision@1 | MRR |
|---|---|---|
| Merged | 10.5 | 0.32 |
| LLM | 19.1 | 0.42 |
| Hybrid | 70.3 | 0.84 |

Best answers are not homogeneous in terms of evidence type and quantity, see Figure 6. As in the broader candidate answer distribution (Figure 5), the best LLM answers have almost double the number of references as the Hybrid answers, on average. Similarly for images, removing an image doesn't necessarily make the answer worse.

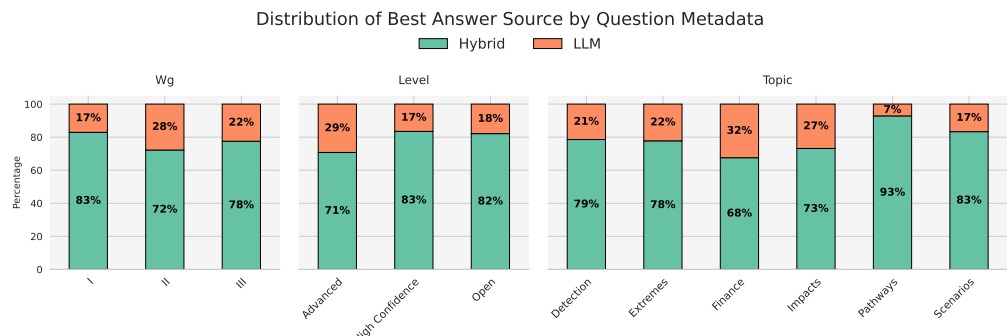

Figure 7: Distribution of best answer source, Hybrid/LLM, by question metadata.

Figure 7 plots the relation between the winning answer source, LLM or Hybrid, and the question metadata: Working Group, Topic, and difficulty level. Here the LLM answer seems to perform better than the hybrid ones particularly on WGII questions (Mitigation, 28%), on more technical question (Advanced, 29%) and Finance (32%) and Impacts (%) topics. While hybrid questions are overwhelmingly preferred on the topic of 'Mitigation Pathways' (97%).

A.4    RUBRICS GENERATION

The following is the model explanation for the generation of the rubric for the question "How does CESM1 Large Ensemble work? What global climate models compose it?"

**Model Explanation:**    "The rubric was designed by systematically analyzing the expert preferences. The core of the analysis involved identifying recurring reasons why one answer was preferred over another. This revealed that while clarity and structure were valued, the most critical differentiators for this specific question were factual accuracy regarding the model's composition and methodology.

The key criteria emerged directly from this analysis:

Explanation of the Large Ensemble Method: This addresses the first part of the question ('How does it work?').

Identification of CESM1 Component Models: This addresses the second part ('What models compose it?').

Accuracy of Technical Details: This criterion was created to capture a crucial point of frequent error identified in the data—specifically, the confusion between the CAM4 and CAM5 atmospheric models. The expert feedback (e.g., preference #52) highlighted this as a key issue. Use of Supporting Visuals: Many answers included diagrams, and these were frequently a deciding factor in preferences, often negatively when an image contradicted the text (e.g., showing a CAM4 diagram for a CAM5-based explanation).

The graph summary was highly useful. It helped identify a top tier of answers (10, 4, 3, 9, 7) and a bottom tier (1, 6, 8), providing a general quality ranking. More importantly, the existence of a single large Strongly Connected Component (SCC) indicated cycles and disagreements among raters. This was critical because it showed that simply counting votes was insufficient. For example, Answer 10 had the most votes but contained a significant factual error (citing CAM4). The cycles suggested that different raters weighted criteria differently (e.g., clarity vs. factual accuracy). This insight led to the creation of a rubric that explicitly separates these dimensions, allowing for a more nuanced assessment that could resolve these apparent contradictions. The Pagerank score was also a helpful secondary indicator, highlighting the strength of Answer 7, which my rubric also identified as a top answer.

If I were to do this again, the process was effective. The combination of qualitative reasons, quantitative vote counts, and graph structure provided a rich dataset to create a robust, data-driven rubric."

Table 3: Elo scores of candidate answers types, grouped by rater type. Scores are shown for all raters combined, as well as for the 'Experts' and 'Advocates' rater subgroups.

| System | All Raters | Rated by Experts | Rated by Advocates |
|---|---|---|---|
| Hybrid curated by Advocates | $1044 \pm 23$ | $1048 \pm 24$ | $1050 \pm 38$ |
| LLM Answer | $1033 \pm 16$ | $1023 \pm 16$ | $1066 \pm 29$ |
| Merged Answer | $1011 \pm 16$ | $997 \pm 19$ | $1062 \pm 26$ |
| Hybrid curated by Experts | $1007 \pm 12$ | $1013 \pm 13$ | $984 \pm 20$ |

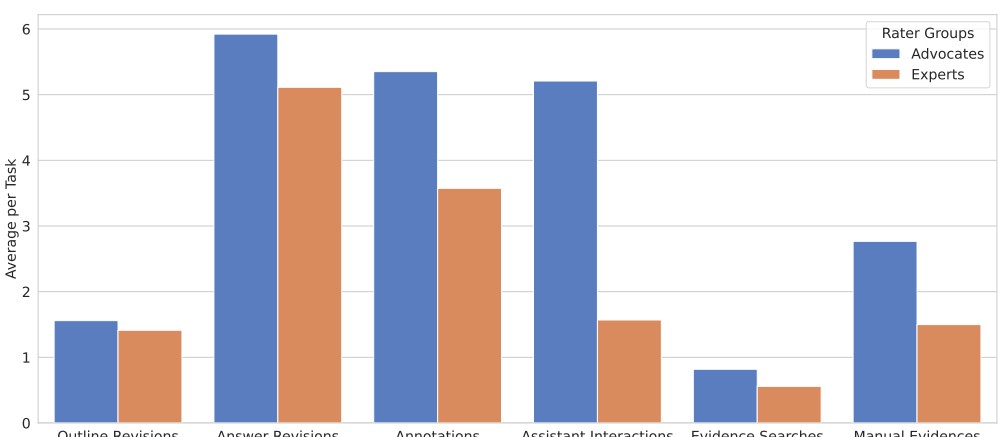

Figure 8: Comparison of Key Interactions in the Editor between Expert groups.

# B HUMAN EXPERTS

## B.1 ADVOCATES

We recruited a group of human raters from the Climate Fresk community. Climate Fresk is an NGO, created by Cedric Ringenbach, to facilitate the understanding of climate issues by the general public.[10] Specifically, the NGO runs workshops were participants work together to understand the basic concepts of climate change and their causal relations. The material used in the workshops is derived from IPCC reports and participants can continue their journey by becoming facilitators and run workshops themselves. The goal behind creating this group was to explore the role of motivation and intrinsic interest in the topic, in combination with the presence of a somewhat uniform background on the topic, if not deep expertise in the academic sense.

Participants in the Advocates group were recruited through an interview, based on their Climate Fresk experience as workshop facilitators. The group consisted of 17 raters, 8 men and 9 women, from various locations: France (8), Morocco (4), Germany (2), Netherlands (2), India (1). In terms of education: 9 had a PhD, 1 is PhD candidate, 5 had Master's, and 2 Bachelor's degrees. All but 2, had read one or two IPCC summary for policy makers. All but 1 had read at least parts of the IPCC Tech summary (6 read it completely). All but 1, at least occasionally, reads scientific papers and all but 3 had contributed to some research (e.g., as data analysts), while 10 had authored or co-authored a scientific paper. Climate Fresk experience was assessed by the number of workshops they facilitated: 1-5 (2), 5-10 (1), 10-20 (5), 20+ (certified trainer) (6), 20+ (certified trainer instructor) (3).

### B.1.1 QUALITY OF ANNOTATIONS

Table 3 shows the scores of model answers, LLM/Merged, or curated either by Advocates (Climate Fresk) or by Experts, and rated by all humans, only Experts or only Advocates. The results show that the Advocates are the best at doing the Phase 1 task as their answers are rated highest by other human participants during Phase 2. Figure 8 compares the key interactions in the Editor between

---

[10]https://climatefresk.org/world/.

Table 4: Elo Score Comparison for the Baseline (CLINB) and ablation Experiments.

| System | CLINB | -Prompt | -Rubrics | -Validation | -Evidence | -Pro+Flash |
|---|---|---|---|---|---|---|
| GPT-5 | $1157 \pm 19$ | $1314 \pm 25$ | $1185 \pm 20$ | $1143 \pm 18$ | $1230 \pm 22$ | $1096 \pm 17$ |
| Claude Opus 4.1 | $1144 \pm 17$ | $1094 \pm 20$ | $1165 \pm 19$ | $1145 \pm 18$ | $1087 \pm 21$ | $1139 \pm 17$ |
| GPT o3 | $1019 \pm 18$ | $1504 \pm 35$ | $1016 \pm 19$ | $1021 \pm 18$ | $1321 \pm 27$ | $1011 \pm 17$ |
| Gemini 2.5 Pro | $968 \pm 18$ | $1037 \pm 21$ | $955 \pm 17$ | $970 \pm 17$ | $1055 \pm 19$ | $962 \pm 16$ |
| Hybrid | $945 \pm 17$ | $567 \pm 31$ | $869 \pm 19$ | $946 \pm 19$ | $749 \pm 24$ | $1010 \pm 18$ |
| Claude Sonnet 4 | $908 \pm 18$ | $740 \pm 21$ | $936 \pm 17$ | $919 \pm 19$ | $777 \pm 22$ | $917 \pm 17$ |
| Gemini 2.5 Flash | $860 \pm 17$ | $744 \pm 23$ | $873 \pm 18$ | $857 \pm 19$ | $781 \pm 22$ | $866 \pm 17$ |

overall and different expert groups. We hypothesize that he Climate Fresk raters are highly motivated so they put in more effort in curating the answer during Phase 1 of the study. They do so through more annotations on the answers, more searches for better evidences, and they manually add more evidences. It seems plausible that through their interactions with the AI-assistant in the Editor, their superior motivation and incentive compensate for the different depth of knowledge. In the human evaluation of Table 1, in only $8\%$ of the battles involving Hybrid answers the human-curated answer is from the Advocates group ($92\%$ are from the Experts). However, if the Hybrid answer wins $13\%$ of the time the answer has been curated by the Advocates.

At the same time, the Advocates are not as good as the Experts at differentiating the quality between different answers. This can be seen by the close ELO scores between answers as rated by Advocates in Table 3. Phase 2 is more difficult to do for participants who don't have the deep knowledge as the Experts do. At the same time, the user interface may be less useful or be more difficult to design for AI assistance, in a side by side.

## C    RESULTS

### C.1    CLINB AUTORATER ABLATIONS

Table 4 Reports the results of various ablation in the model-assessment procedure, by varying the .

Table 4 Reports the results of various ablation in the model-assessment procedure. The baseline (CLINB) uses the full prompt: guidelines, scientist-reviewed rubrics, evidence handling instructions, step-by-step instructions, and gemini-2.5-pro as the rater.

**-Prompt**    In this configuration we replace the CLINB prompt with a simple vanilla prompt requesting the LLM to choose the better answer. The results change substantially: the human-curated answers get the lower score observed in our experiments. Claude Opus 4.1 experiences also a regression. Gemini 2.5 Pro and the OpenAI GPT-5 get a boost, with OpenAI o3's ELO score increasing by $50\%$. The LLM-as-a-Judge paradigm is known to be susceptible to biases, including favoring style over substance Gudibande et al. (2024) and AI-AI bias (Panickssery et al., 2024). This probably reveal the underlying bias and stylistic preferences of the judge model and demonstrate the the effectiveness of the CLINB autorater.

**-Rubrics**    Here we remove the question-specific Rubrics components from the CLINB prompt (see Appendix D.3 for the full prompt). The ELO scores are close to those obtained with the full CLINB autorater. However, Hybrid answers drops to last place and the score of the Gemini 2.5 Flash improve to overtake the human-curated answers. Notice that Gemini 2.5 Flash and the Hybrid answers are the data from which the rubrics were induced, vias Phase 1 and Phase 2. This shows that without the question-specific rubrics the model cannot use the detailed information about their relative strengths and weaknesses. The result suggests that data-derived rubrics require more data to be representative of the common concepts, pitfalls and misconception of multiple models and possibly require an adaptive approach where they are constantly updated.

**-Validation**    uses rubrics generated by Gemini 2.5 Pro. These rubrics are generated using experts curated answers and rankings so they already include a lot of expert knowledge. These are the starting rubrics the scientists review and revise into the final rubrics. The scientists were generally very impressed by these rubrics and only made minor improvements during their review so unsurprisingly, the performance of *-Validation* is the same as CLINB within the confidence interval.

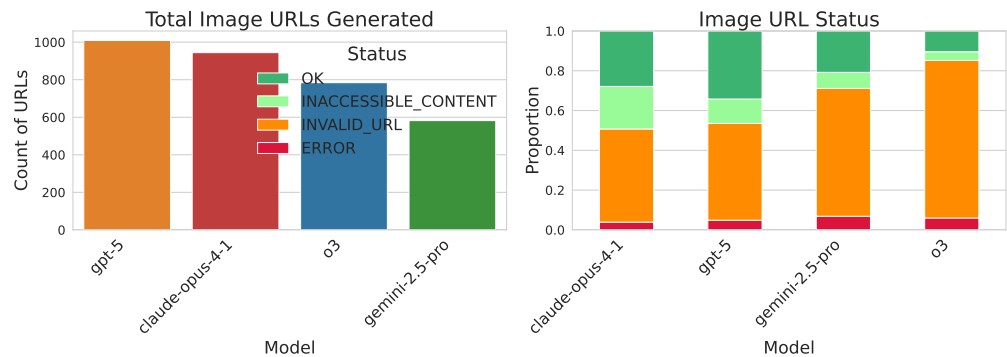

Figure 9: Image URL Status Breakdown for Answers with Mandatory Image

**-Evidence** removes the evidence penalty. This is the most crucial component of CLINB because without it, the autorater cannot tell a valid evidence from an invalid one. It's unable to discount key facts that are unsupported correctly and the scores tend to revert towards the baseline (-Prompt).

**-Pro+Flash** swaps out the rating model to Gemini 2.5 Flash. The weaker rating model is not as good at linking multiple pieces of information together from the answer, to the rubrics, and to the associated evidence validity. It is often unable to discount facts backed by invalid evidence so the overall ranking of the models do not agree with CLINB.

### C.2 VISUAL EVIDENCE

Most models choose to omit the image in the answer when it is described as optional. When the request to include an image in the response is made "mandatory" most of the image links are hallucinated (Figure 9).

### C.3 QUESTION CATEGORY AND TOPICS

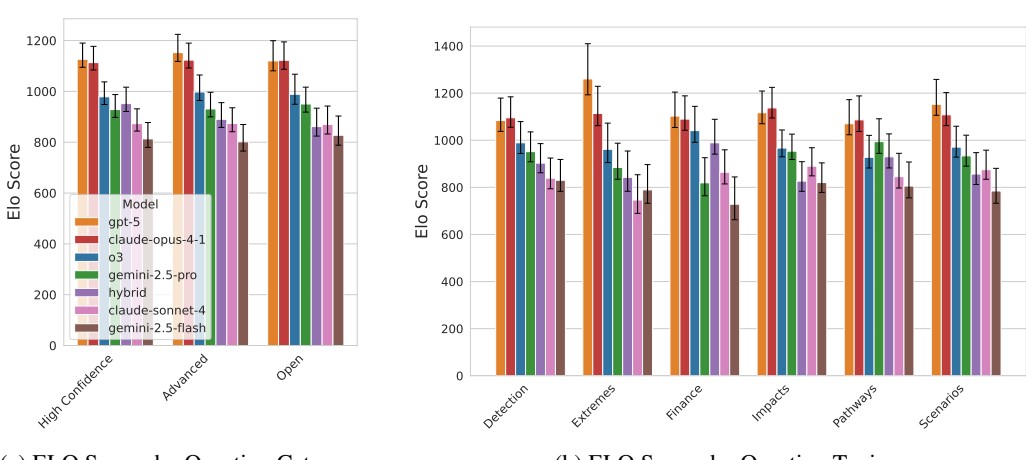

(a) ELO Scores by Question Category      (b) ELO Scores by Question Topic

In Figure 10a, we look at how well the models answer different questions categories according to the CLINB autorater: High Confidence, Advanced, and Open. The overall rankings of the models are similar between the 3 groups, and close to the general results, indicating that question complexity does not have a marked effect on answer quality. There are slight variations in performance for the models in the middle. Hybrid answers are slightly better for the High Confidence questions, suggesting humans felt more comfortable improving over the model generated answer for these questions. Perhaps it's because there is a high degree of scientific consensus here.

In Figure 10b, we consider how well the models can answer questions on different topics: Detection, Extremes, Finance, Impacts, Pathways, and Scenarios. There is more variation between the different

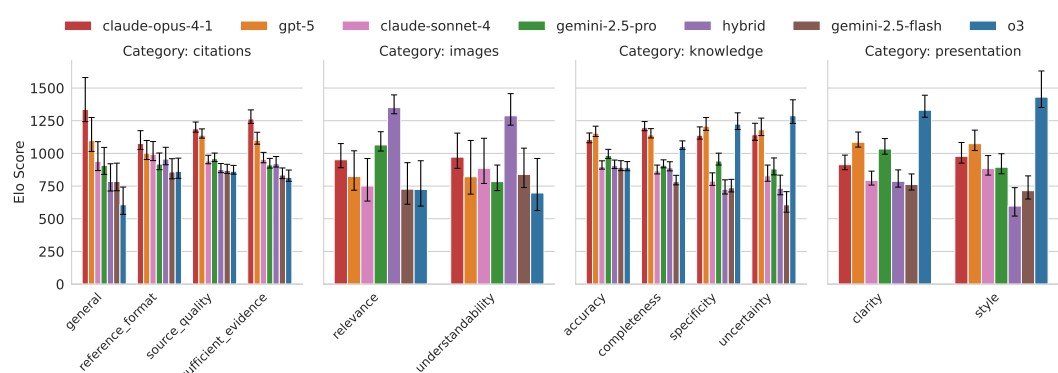

Figure 11: ELO Ratings by Evaluation Dimensions according to CLINB

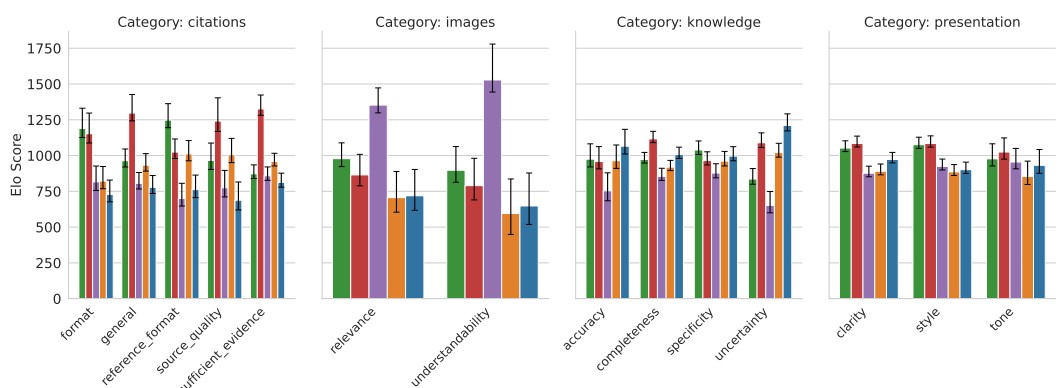

Figure 12: ELO Ratings by Evaluation Dimensions according to Experts

systems when the data is sliced this way. Hybrid answers performed relatively well on "Climate Finance and Risks". Perhaps this means there is higher uncertainty on this topic in the models' parametric knowledge.

### C.4 DETAILED EVALUATION DIMENSIONS

Figure 11 reports the performance, according to the CLINB autorater, on a more specific layer of the evaluation dimensions. Claude Opus 4.1 and GPT-5 are head to head on the knowledge dimensions, Claude Opus 4.1 is best on citations, Hybrid answers have the best images, while OpenAI o3 wins in presentation. Figure 12 shows the same performance breakdown evaluated by Experts. The experts are not as confident as the CLINB autorater at judging the knowledge dimensions. We hypothesize that this is due to the fact that exhaustive checking of all technical answer details and nuances is too time confusing for advanced answers. The manual pairwise results show more confidence and resolution on the evidence, images and references, which are easier to check at least superficially.

## D PROMPTS

We report here verbatim all prompts used in our experiments. Prompts are formatted in Markdown.

### D.1 GUIDELINES

These guidelines are used both in the autorater prompt and in the system answer prompt, for consistency. fontsize

```
Unless there are significant presentation flaws, disregard stylistic
↪   aspects; our goal is to evaluate an answer primarily on its
↪   epistemological quality.
```

```
1188  In case of ambiguity or underspecification, the question should be
1189  ↪   interpreted as relevant and specific to the topic of
1190  ↪   **climate change**.
1191
1192  The answers must adhere strictly to the following guidelines and
1193  ↪   diagnostic dimensions.
1194
1195  * **Tone:** Answers must maintain a strictly
1196  ↪   **neutral, balanced, and unbiased** tone. Avoid subjective language,
1197  ↪   sensationalism, alarmism, exaggeration, or any emotionally loaded
1198  ↪   phrasing. Focus on objective assessment.
       * **Accuracy:** All information provided must be **accurate** according
1199  ↪   to current scientific understanding and adequate scientific
1200  ↪   standards. Statements that present scientific findings out of
1201  ↪   context, are self-contradictory or rely on anecdotal evidence must be
       ↪   avoided.
1202  * **Specificity:** The answer must address **only and exclusively** what
1203  ↪   the questions explicitly and directly asks. If elaboration is needed,
1204  ↪   it should deepen the detail on the specific topic rather than
1205  ↪   broadening the scope to related but irrelevant information. Strictly
       ↪   adhere to any specified context (e.g., temporal or geographical).
1206  * **Completeness:** The answer must address **all parts** of the question
1207  ↪   thoroughly. Not omit important aspects. Provide sufficient detail,
1208  ↪   including relevant numbers and statistics where appropriate.
1209  ↪   Acknowledge established scientific knowledge, relevant findings, and
       ↪   different significant perspectives on the question where applicable.
1210  * **Uncertainty:** The answer must appropriately convey the
1211  ↪   **degree of uncertainty** associated with the information presented,
1212  ↪   especially regarding scientific findings or projections. Significant
1213  ↪   contradicting evidence or alternative interpretations, if they exist
       ↪   within the scientific community, must be mentioned.
1214  * **Sources:**
1215      * **Every key point, statement, statistic, image or piece of data**
1216      ↪   presented in the answer must be supported by a traceable citation
1217      ↪   linked to the reference section.
1218      * Evidence from **first-hand academic sources**, particularly
           ↪   peer-reviewed articles must be prioritized.
1219      * The answer must rely only on **trustworthy and authoritative**
1220      ↪   sources and favor **up-to-date evidence** when newer reliable
1221      ↪   data or findings are available.
1222      * All references must always include **valid URLs**.
1223
```

## D.2    SYSTEM PROMPT

This is the prompt used to query LLMs to be evaluated. Notice that the prompt doesn't specify *how* an answer should be constructed. We argue that it should be up to the system's *intelligence*, i.e., its planning, agentic, self-critiquing, reasoning skills to figure out how to research and compose an adequate response. fontsize

```
1230  **Task:** Carry out the necessary research and answer the question below.
1231
1232  **Contextual Interpretation:** If the question is ambiguous or
1233  ↪   underspecified, interpret it as being relevant and specific to the
1234  ↪   topic of **climate change**.
1235  **Input Question:**
1236  \{user\_question\}
1237
1238  **Output Requirements:**
1239
1240  Your answer must strictly adhere to the following format and quality
      ↪   guidelines:
1241
      **1. Format:**
```

```
1242
1243    * **Main Body:**
1244        * Maximum length:
1245        ↪  **\{MAX\_NUM\_WORDS\_IN\_BENCHMARK\_ANSWER\} words** (counted by
1246        ↪  splitting text at whitespaces). Content exceeding this limit will
1247        ↪  be disregarded.
1248        * Must be formatted using **Markdown**.

1249    * **Reference Section:**
1250        * Must list **all** sources cited in the main body.
1251        * Citations within the text must use the **IEEE style** (e.g., `[1]`,
1252        ↪  `[2]`).
1253        * Sources must be numbered sequentially in the order they first
1254        ↪  appear in the text.
1255        * Each reference must include a **valid URL link** to the source.
1256        * Non-academic sources (e.g., web pages, reports) must at least
1257        ↪  include the title and the source name (e.g., organization,
1258        ↪  website). Academic sources should follow standard IEEE citation
1259        ↪  format including authors, year, title, publication venue etc.,
1260        ↪  plus the URL.

1260    * **Images (Optional):**
1261        * Images can greatly enhance the clarity and effectiveness of the
1262        ↪  answer. Thus, answers may include images **but only** if they
1263        ↪  effectively summarize quantitative data, results, describe
1264        ↪  processes, or significantly aid understanding.
1265        * Each image must be accompanied by a **caption** describing its
1266        ↪  content.
1267        * The caption must **cite the source** of the image, following the
1268        ↪  same IEEE citation style used in the text, linking to the
1269        ↪  corresponding entry in the reference section.
1270        * Each image must include a **valid URL link** directly to the image
1271        ↪  file itself if possible, or to the page containing the image.
1272        * Use Markdown for image embedding (`![alt text](Image URL)`). Don't
1273        ↪  use any other image embedding syntax.

1272  **2. Quality Guidelines:**

1274  The following guidelines must be strictly followed when generating the
1275  ↪  answer:
1276  \{GUIDELINES\}

1277  **Proceed to generate the answer based *only* on these instructions.**
1278
1279
```

## D.3  EVALUATION PROMPT

This is the prompt CLINB uses for pairwise evaluation of answers. fontsize

```
# Task Description

You are an expert climate scientist and internationally known author.
↪  Your task is to compare two answers to the same question and provide
↪  a detailed assessment of the answers along with a final decision on
↪  which answer is better.

You task involves carefully comparing two answers to the same question
↪  using the materials listed below:
* The **Question**, marked within <START QUESTION> and <END QUESTION> to
↪  be answered.
* **Answer 1**, marked within <START ANSWER 1> and <END ANSWER 1>, the
↪  first answer.
* **Answer 2**, marked within <START ANSWER 2> and <END ANSWER 2>, the
↪  second answer.
```

```
1296    * **Instructions**, marked within <START INSTRUCTIONS> and <END
1297    ↪  INSTRUCTIONS>, the general instructions on how to perform the task,
1298    ↪  decide which answer is better, and why, in a principled and
1299    ↪  structured way.
1300    * **Guidelines**, marked within <START GUIDELINES> and <END GUIDELINES>,
1301    ↪  the primary evaluation dimensions.
1302    * **Grader's Cheat Sheet**, marked within <START CHEAT SHEET> and <END
1303    ↪  CHEAT SHEET>, the question-specific background information that is
1304    ↪  necessary for an accurate and consistent evaluation.
1305    * **Question-specific Grading Rubrics**, marked within <START RUBRICS>
1306    ↪  and <END RUBRICS>, the answer grading standards that are specific to
1307    ↪  the question.
1308    * **Shared Grading Rubrics**, marked within <START SHARED RUBRICS> and
1309    ↪  <END SHARED RUBRICS>, the grading standards for the answers, that are
1310    ↪  shared across multiple questions.
1311    * **Supplemental Materials**, marked within <START SUPPLEMENTAL
        ↪  MATERIALS> and <END SUPPLEMENTAL MATERIALS>, the supplemental
        ↪  materials that are used for the evaluation.

1312    # Main Task Materials
1313    <START QUESTION>
1314    [The question is inserted here]
1315    <END QUESTION>
1316
1317    <START ANSWER 1>
1318    [answer 1 is inserted here]
        <END ANSWER 1>
1319
1320    <START ANSWER 2>
1321    [answer 2 is inserted here]
        <END ANSWER 2>
1322
1323    <START INSTRUCTIONS>
1324    # Step-by-step Instructions
1325
1326    ## **Step 1: Analyze the Question and the Answer Contents**
1327
        First, get familiar with the question and content of both answers.
1328
1329    ## **Step 2: Analyze the Grading Rubric**
1330
1331    The question-specific Grading Rubric is marked within <START RUBRICS> and
        ↪  <END RUBRICS>.
1332    The shared Grading Rubric is marked within <START SHARED RUBRICS> and
1333    ↪  <END SHARED RUBRICS>.
1334
1335    The Grading Rubric has been created along the following core assessment
        ↪  criteria:
1336    * **Scientific Accuracy and Depth:** Correctness and thoroughness of the
1337    ↪  climate science concepts.
1338    * **Clarity of Argument and Structure:** Logical flow, coherence, and
1339    ↪  clear writing.
1340    * **Use of Evidence (Images & Data):** Effective and accurate integration
1341    ↪  of images to support the argument.
1342    * **Quality of Citations and References:** Appropriate use of
        ↪  high-quality sources.
1343    * **Adherence to the 'Guidelines':** How well the answer follows the
1344    ↪  specified quality guidelines.
1345
1346    In addition to the question-specific Grading Rubric, you will also be
        ↪  provided with the shared Grading Rubric, marked within <START SHARED
1347    ↪  RUBRICS> and <END SHARED RUBRICS>.
1348    This shared Grading Rubric is an additional set of quality criteria for
1349    ↪  the answers. However pay most attention to the question-specific
        ↪  Grading Rubric and the Grader's Cheat Sheet.
```

You should analyze both the question-specific Grading Rubric and the
↪   shared Grading Rubric, with an
↪   **emphasis on the question-specific Grading Rubric**.

## **Step 3: Analyze the Grader's Cheat Sheet**

The Grader's Cheat Sheet is marked within <START CHEAT SHEET> and <END
↪   CHEAT SHEET>.
It can help you apply the rubric quickly and consistently.
It includes the following sections:
* **Key Scientific Concepts to Look For:** A bulleted list of the
↪   essential scientific points a top answer must include.
* **Common Pitfalls & Misconceptions:** A list of frequent errors or
↪   weaker arguments.
* **Checklist for Evaluation of Images and References:** 2-3 questions
↪   that guide the assessment of images and references.

## **Step 4: Analyze the Answer Guidelines**

Analyze the provided Answer Guidelines, which is a set of quality
↪   criteria for answers.

## **Step 5: Analyze and Grade Each Answer Individually.**

Meticulously analyze each answer's content with respect to the above
↪   Grading Rubrics and the Grader's Cheat Sheet.
Make this analysis in depth and as precise as possible and reflect on
↪   your choices.
Then go through the rubrics one by one and, for each one, provide an
↪   assessment and a rating for both answers.
In addition, for each criterion, compare the two answers and decide if
↪   one answer is better than the other.
It is possible for both answer to have the same score but one can still
↪   have an edge over the other one.
Use 'answer_1_preferred' to indicate that answer_1 is better than
↪   answer_2 and 'answer_2_preferred' to indicate that answer_2 is better
↪   than answer_1.
If there is no clear winner, declare a 'tie'.

Consider the following guidelines when rating the answers:
* **Focus on the Question-Specific Grading Rubric:** Prioritize the
↪   question-specific Grading Rubric and the Grader's Cheat Sheet over
↪   the shared Grading Rubric and the Answer Guidelines.
* **Provide Detailed and Scientific Rationales:** Provide a detailed and
↪   scientific rationale for your ratings, including supporting evidence
↪   from the answer and the Grading Rubric.
* **Be Critical:** Be thorough and rigorous in your analysis and ratings.

## **Step 6: Make a Final Decision**

Finally, make a final decision on which answer is better based on your
↪   previous analysis and the available materials.
Give a short explanation of what aspects of the answers are the main
↪   drivers for your decision.

Use the Supplemental Materials to determine if any penalty should be
↪   applied to the answer.

Explain why the preferred answer is better than the other one based on
↪   the grading materials.
If an answer contains images pay particular attention to them and how
↪   they contibute to the answer.

## Guidelines

```
In your assessment, follow strictly the evaluation dimensions explained
↪ in the guidelines below:
<START GUIDELINES>
[The guidelines (above) are inserted here]
<END GUIDELINES>

## Grader's Cheat Sheet
Use the Cheat Sheet to help you understand the question and the
↪ background information necessary to formulate a calibrated and
↪ consistent assessment.
<START CHEAT SHEET>
[Question specific cheat sheet is inserted here]
<END CHEAT SHEET>

## Specific Grading Rubrics
Use the Grading Rubrics to implement consistent and accurate grading
↪ standards.
<START RUBRICS>
[Question specific rubrics are inserted here]
<END RUBRICS>

## Shared Grading Rubrics
Use the Shared Grading Rubrics to implement consistent and accurate
↪ grading standards.
<START SHARED RUBRICS>
[
  {
    "criterion": "Tone",
    "description": "Evaluates the neutrality, balance, and objectivity of
    ↪ the language used in the answer.",
    "requirements_score_9_10": "The answer maintains a consistently
    ↪ neutral, balanced, and unbiased tone throughout. It is entirely
    ↪ free of subjective language, sensationalism, alarmism,
    ↪ exaggeration, or emotionally loaded phrasing. The focus is purely
    ↪ on objective, factual assessment.",
    "requirements_score_7_8": "The tone is overwhelmingly neutral and
    ↪ objective, but there may be very minor, isolated instances of
    ↪ slightly subjective or subtly leading language that do not
    ↪ significantly impact the overall balance of the answer.",
    "requirements_score_5_6": "The answer attempts to maintain a neutral
    ↪ tone, but there are noticeable lapses into subjective, slightly
    ↪ exaggerated, or emotionally tinged language in several places,
    ↪ which moderately detracts from the overall objectivity.",
    "requirements_score_1_4": "The tone is frequently and significantly
    ↪ biased, subjective, sensationalist, or alarmist. Emotionally
    ↪ loaded language is common and severely compromises the
    ↪ objectivity and neutrality of the answer.",
    "requirements_score_0": "The tone is entirely inappropriate,
    ↪ propagandistic, or completely fails to adopt a factual and
    ↪ objective stance. The answer is unrateable on its epistemological
    ↪ merits due to its tone."
  },
  {
    "criterion": "Accuracy",
    "description": "Assesses the factual correctness of the information
    ↪ provided, its contextual relevance, and its basis in current
    ↪ scientific understanding.",
    "requirements_score_9_10": "All information presented is entirely
    ↪ accurate according to current, mainstream scientific
    ↪ understanding. All scientific findings are presented in their
    ↪ proper context, without contradictions or reliance on anecdotal
    ↪ evidence. The information reflects the highest scientific
    ↪ standards.",
```

```
      "requirements_score_7_8": "The vast majority of the information is
      ↪   accurate and well-contextualized. There may be a minor factual
      ↪   error or a slightly out-of-context statement that does not
      ↪   undermine the core argument or the overall correctness of the
      ↪   answer.",
      "requirements_score_5_6": "The answer contains some accurate
      ↪   information but also includes several noticeable factual errors,
      ↪   misinterpretations, or statements presented out of their proper
      ↪   scientific context. The answer may contain minor
      ↪   self-contradictions.",
      "requirements_score_1_4": "The answer contains significant, numerous,
      ↪   or fundamental factual errors. Information is frequently
      ↪   presented out of context, is self-contradictory, or relies on
      ↪   discredited science or anecdotal evidence. The core claims are
      ↪   scientifically unsound.",
      "requirements_score_0": "The answer is completely factually
      ↪   incorrect, based on pseudoscience or misinformation, or provides
      ↪   no verifiable information."
    },
    {
      "criterion": "Specificity",
      "description": "Evaluates how strictly the answer adheres to the
      ↪   scope of the question, including any explicit constraints (e.g.,
      ↪   temporal or geographical).",
      "requirements_score_9_10": "The answer addresses only and exclusively
      ↪   the explicit question asked. It does not introduce related but
      ↪   irrelevant topics. Any elaboration serves only to deepen the
      ↪   detail on the specific topic. All specified contexts (e.g.,
      ↪   timeframe, location) are strictly respected.",
      "requirements_score_7_8": "The answer is highly focused on the
      ↪   question but may include a minor, brief deviation into a
      ↪   tangentially related topic that does not significantly detract
      ↪   from the main focus. It largely respects all specified
      ↪   contexts.",
      "requirements_score_5_6": "The answer addresses the core question but
      ↪   also includes a noticeable amount of irrelevant information or
      ↪   broadens the scope beyond what was asked. It may partially
      ↪   neglect or misinterpret a specified context.",
      "requirements_score_1_4": "The answer significantly deviates from the
      ↪   question, focusing largely on related but irrelevant topics. The
      ↪   core question is only superficially addressed or misunderstood.
      ↪   Specified contexts are largely ignored.",
      "requirements_score_0": "The answer completely fails to address the
      ↪   question asked and is entirely off-topic."
    },
    {
      "criterion": "Completeness",
      "description": "Assesses whether the answer thoroughly addresses all
      ↪   parts of the question, providing sufficient detail and
      ↪   acknowledging all relevant perspectives.",
      "requirements_score_9_10": "The answer comprehensively addresses all
      ↪   parts of the question, leaving no significant aspect unexamined.
      ↪   It provides a thorough level of detail, including relevant and
      ↪   well-chosen data and statistics. It accurately represents the
      ↪   established scientific consensus and acknowledges different
      ↪   significant scientific perspectives where applicable.",
      "requirements_score_7_8": "The answer addresses all major parts of
      ↪   the question but may omit a minor aspect or provide slightly less
      ↪   detail than would be ideal in one area. The overall picture is
      ↪   still robust and well-supported.",
      "requirements_score_5_6": "The answer addresses the main parts of the
      ↪   question but omits some important aspects, fails to provide
      ↪   sufficient detail, or lacks nuance.",
```

```
    "requirements_score_1_4": "The answer is notably incomplete, omitting
    ↪  major parts of the question or providing only superficial,
    ↪  cursory information. Key details, data, or established scientific
    ↪  perspectives are missing.",
    "requirements_score_0": "The answer is a fragment or completely fails
    ↪  to provide a substantive response to any part of the question."
  },
  {
    "criterion": "Sources",
    "description": "Assesses the quality, appropriateness, and proper
    ↪  citation of the sources used to support the answer.",
    "requirements_score_9_10": "Every key point, statement, statistic,
    ↪  image or piece of data is meticulously supported by a traceable
    ↪  citation linked to the reference section. Sources are
    ↪  overwhelmingly first-hand, peer-reviewed academic literature or
    ↪  reports from major scientific bodies (e.g., IPCC, Nature, WHO).
    ↪  All sources are authoritative, up-to-date, and include a valid,
    ↪  working URL.",
    "requirements_score_7_8": "Most key points are properly cited with
    ↪  high-quality, authoritative sources. There may be a few minor
    ↪  statements lacking a direct citation, or a small number of
    ↪  citations may be to high-quality secondary sources (e.g.,
    ↪  reputable scientific journalism or a good Wikipedia article)
    ↪  instead of primary literature. All URLs are valid.",
    "requirements_score_5_6": "Citations are present but inconsistent;
    ↪  some key claims lack support. The answer relies significantly on
    ↪  secondary sources, news articles, or institutional websites
    ↪  rather than primary academic literature. Some sources may be
    ↪  slightly outdated.",
    "requirements_score_1_4": "Citations are largely missing, incorrect,
    ↪  or link to unreliable sources (e.g., blogs, opinion pieces,
    ↪  non-scientific websites). There is little to no use of
    ↪  peer-reviewed literature. Many key claims are unsupported.",
    "requirements_score_0": "The answer provides no sources or citations
    ↪  whatsoever, or the sources provided are entirely inappropriate
    ↪  and untrustworthy."
  }
]
<END SHARED RUBRICS>
```

## Supplemental Materials

```
Use the Supplemental Materials to determine if the referenced images and
↪  URLs are valid and accessible. If an URL is not explicitly included
↪  in the Supplemental Materials, assume it is valid. The answer should
↪  be penalized if it contains invalid images or URLs.
If an image is indicated as important for answering the question but it
↪  is missing in the answer being evaluated, the answer should also be
↪  penalized.
Assume that references are important for answering the question. If there
↪  are no references in the answer being evaluated, the answer should be
↪  penalized. Also, the answer with more valid references should be
↪  rewarded.

If an answer contains invalid references, you must assume that the facts
↪  supported by those references are missing from the answer as they
↪  cannot be verified.
If key facts in the answer are not supported by a valid reference, you
↪  must also assume those facts are missing from the answer as they
↪  cannot be verified.
This will affect how you evaluate the Completeness, Accuracy, and
↪  Specificity of the answer.
<START SUPPLEMENTAL MATERIALS>
[Answer specific supplemental materials are inserted here. These includes
↪  the results of checking the validity of each referenced URL.]
<END SUPPLEMENTAL MATERIALS>
```

```
1566   <END INSTRUCTIONS>
1567
1568   Output format is JSON:
1569   {
1570     "specific_rubric_ratings: [  # The specific rubric ratings as given
1571     ↪   above between <START RUBRICS> and <END RUBRICS>.
1572         {
1573           criterion: str # The name of the criterion.
1574           eval_answer_1: str  # Short explanation of how answer 1 scores
1575           ↪   against the rubric.
1576           rating_answer_1: float  # Float (with single decimal space) from
1577           ↪   0 to 10
1578           eval_answer_2: str  # Short explanation of how answer 2 scores
1579           ↪   against the rubric.
1580           rating_answer_2: float  # Float (with single decimal space) from
1581           ↪   0 to 10
1582           pair_eval_rationale: str # Rationale for the pair evaluation of
1583           ↪   answer 1 vs answer 2.
1584           pair_eval: str  # Either "answer_1_preferred",
1585           ↪   "answer_2_preferred", or "tie"
1586         },
1587     ],
1588     "shared_rubric_ratings": [  # The shared rubric ratings as given above
1589     ↪   between <START SHARED RUBRICS> and <END SHARED RUBRICS>.
1590         {
1591           criterion: str # The name of the criterion.
1592           eval_answer_1: str  # Short explanation of how answer 1 scores
1593           ↪   against the rubric.
1594           rating_answer_1: float  # Float (with single decimal space) from
1595           ↪   0 to 10
1596           eval_answer_2: str  # Short explanation of how answer 2 scores
1597           ↪   against the rubric.
1598           rating_answer_2: float  # Float (with single decimal space) from
1599           ↪   0 to 10
1600           pair_eval_rationale: str # Rationale for the pair evaluation of
1601           ↪   answer 1 vs answer 2.
1602           pair_eval: str  # Either "answer_1_preferred",
1603           ↪   "answer_2_preferred", or "tie"
1604         },
1605     ],
1606     "decision": str, # Either "answer_1_preferred", "answer_2_preferred",
1607     ↪   no ties allowed here.
1608     "explanation": str # The rationale behind the decision.
1609   }
1610   Output:
```

## E  INFORMAL CHARACTERIZATIONS

**GPT-5's** answers are rated as accurate, complete, specific, nuanced, up to date, and of great scientific depth. Regarding presentation they are logically organized and often include some advanced concepts, that others models omit. They tend to be text-heavy though and light on helpful formatting.

**OpenAI o3** is praised for scientific nuance and completeness. OpenAI o3 is more likely to give details, cite numbers and direct facts. On the flipside this opens up the answer to being factually wrong for each of these facts. The model also seems to have more problems with hallucinated references than other models. Another strength is presentation, in particular using tables, formatting, and a clear logical flow. As an example for accessible formatting, for the question "Can you detail the assumptions in the five different climate scenario SSP1-1.9, SSP1-2.6, SSP2-4.5, SSP3-7.0 and SSP5-8.5.", OpenAI o3 produced a clear table allowing for easy side-by-side comparison of the different scenarios. This was preferred over the narrative styles of the other models, particularly Claude Opus 4.1.

Our analysis shows that **Claude Opus 4.1** answers not only rate among the best for scientific depth, comprehensiveness, specificity, and nuance but also provide verifiable data and quantitative detail,

backed by extensive references. The main weakness seems to lie in interpretation of the question: sometimes Claude Opus 4.1 misinterprets the question and answers something else. For example, for a question about impact attribution studies of extreme weather events, Claude Opus 4.1 fails to define *impact attribution*, i.e. how climate change influences the *outcomes* of an extreme weather event, and instead produces a comprehensive answer on event attribution, i.e. quantifying how climate change affects the likelihood of the extreme weather event *itself*.

According to the analysis, **Gemini 2.5 Pro's** strength lies in providing reliable and accurate answers backed by high-quality sources. The model seems to be good at correctly interpreting the question, identifying the core scientific principles and giving a direct and correct answer. Compared to other models, Gemini 2.5 Pro answers tend to be less extensive and can lack some nuance, completeness, or concrete numbers.

Finally, the main strengths for the **Hybrid** answers are verifiability and presentation. These answers often win because all references can be resolved, in contrast to other systems, as humans are immune to hallucinating URLs. In addition, most Hybrid answers include helpful images and graphs (see Figure 3, see appendix F). On the flip side, Hybrid answers can be superficial and lack some concrete numbers or examples (also appendix F).

# F EXAMPLES

In this section we investigate an example of a disagreement between the human preferences and our rubrics based autorater. The goal is to gain a high-level understanding of potential biases humans or prompted LLMs might exhibit, not to second guess the human rating to align with the automatic ones. We believe that there is value in having a multifaceted assessment and thus focus on the correctness of the justification over the individual preferences.

## F.1 EXAMPLE 1: WHAT KIND OF DATA AND MEASUREMENT TOOL IS BEST FOR CLIMATE RISK MODELING AND DISCLOSURE

**Question**   *What kind of data and measurement tool is best for climate risk modeling and disclosure?*

**Summary of Verdicts**   Scientists preferred the model answer, citing superior presentation clarity and style. The Autorater preferred the Hybrid (Human + Gemini Flash) answer.

**Analysis**   The Autorater's preference for the Hybrid answer highlights the importance of substantive completeness as defined by the expert-curated rubric.

1. **Addressing Uncertainty and Challenges:** The rubric explicitly required a sophisticated discussion of the inherent challenges and uncertainties in climate risk modeling. The Autorater found the Hybrid answer vastly superior on this dimension. The human expert intervention ensured this critical aspect was covered comprehensively, while the model response addressed them only superficially.

2. **Citation and Image Quality:** The Autorater penalized the model answer for including invalid references supporting critical frameworks (NGFS and IFRS S2) and for providing a broken image link.

This divergence demonstrates how the Autorater, guided by the rubric, prioritizes essential scientific content—such as the acknowledgment of uncertainty—over stylistic clarity, mitigating potential presentation bias in evaluation. Furthermore, this case illustrates the value of the expert-in-the-loop approach, where human curation ensured substantive completeness even when starting from a less capable base model (Gemini 2.5 Flash) compared to a stronger autonomous model.

### F.1.1 RUBRICS

See Tables tables 5 to 7 for the specific and shared rubrics used.

Table 5: Question specific rubrics, reviewed and edited by *scientists*.

| Question | What kind of data and measurement tool is best for climate risk modeling and disclosure? |
|---|---|

**Risk Categorization and Scope**

| | |
|---|---|
| description | Assesses the clarity and accuracy of the distinction between physical and transition risks, and the breadth of the discussion to cover various sectors, not just finance. |
| score 9-10 | Clearly and accurately defines and distinguishes between physical risks (both acute and chronic) and transition risks (policy, technology, market). The discussion is comprehensive and not limited to a single sector like finance. |
| score 7-8 | Correctly defines and distinguishes physical and transition risks, but may lack detail on their sub-types. The scope is generally appropriate but might lean heavily towards one sector. |
| score 5-6 | Mentions both risk types, but the distinction is unclear or the definitions are superficial. The scope is likely too narrow (e.g., only finance). |
| score 1-4 | Fails to clearly distinguish between risk types, discusses only one type, or the definitions are incorrect. |
| score 0 | Does not address the different categories of climate risk. |

**Specificity of Data and Tools**

| | |
|---|---|
| description | Evaluates the level of detail in describing the specific data types, models, and analytical techniques for both physical and transition risk modeling. |
| score 9-10 | Provides a rich, specific list of data types and tools for both risk categories. Mentions specific model types (e.g., GCMs/RCMs, catastrophe models), techniques (e.g., downscaling, scenario analysis), and data sources (e.g., geospatial data, policy trackers). |
| score 7-8 | Lists relevant data types and tools for both risk categories but with less specific detail. For example, might mention 'climate models' without specifying GCMs, or 'financial models' without mentioning scenario analysis. |
| score 5-6 | Describes data and tools in general terms (e.g., 'climate data,' 'modeling'). The link between specific risks and the corresponding tools is weak or absent. |
| score 1-4 | Mentions data or tools, but the description is vague, inaccurate, or highly incomplete. |
| score 0 | Fails to mention relevant data or measurement tools. |

**Discussion of Frameworks and Standards**

| | |
|---|---|
| description | Assesses the inclusion and explanation of key industry and regulatory frameworks that guide climate risk disclosure and assessment. |
| score 9-10 | Explicitly names and accurately describes the role of key disclosure frameworks (e.g., TCFD, ISSB) and may also mention data standards (e.g., GHG Protocol). Effectively uses or describes a conceptual assessment framework to structure the process. |
| score 7-8 | Names relevant frameworks (e.g., TCFD) but provides a limited explanation of their role or significance. May include a conceptual framework diagram but with minimal integration into the text. |
| score 5-6 | Mentions frameworks in passing or lists them without any explanation of their purpose. |
| score 1-4 | Fails to mention key frameworks or shows a misunderstanding of their purpose. |
| score 0 | No mention of any frameworks or standards. |

**Nuance and Acknowledgment of Challenges**

| | |
|---|---|
| description | Evaluates the answer's acknowledgment of the inherent complexities, uncertainties, and limitations in climate risk modeling and disclosure. |
| score 9-10 | Provides a sophisticated discussion of key challenges, such as data availability and granularity, model limitations, inherent uncertainty in climate projections, and issues with the transparency and comparability of tools. |
| score 7-8 | Acknowledges that uncertainties and challenges exist, but the discussion is less detailed. May mention 'data gaps' or 'model uncertainty' without significant elaboration. |
| score 5-6 | Makes a brief, generic statement about uncertainty without connecting it to specific aspects of climate risk modeling. |
| score 1-4 | Ignores the topic of uncertainty and challenges, presenting information as if it were perfectly known and straightforward. |
| score 0 | The answer is too brief or irrelevant to assess this criterion. |

Table 6: Shared (non question specific) rubrics used for evaluation (Part 1).

**Tone**

| | |
|---|---|
| description | Evaluates the neutrality, balance, and objectivity of the language used in the answer. |
| score 9-10 | The answer maintains a consistently neutral, balanced, and unbiased tone throughout. It is entirely free of subjective language, sensationalism, alarmism, exaggeration, or emotionally loaded phrasing. The focus is purely on objective, factual assessment. |
| score 7-8 | The tone is overwhelmingly neutral and objective, but there may be very minor, isolated instances of slightly subjective or subtly leading language that do not significantly impact the overall balance of the answer. |
| score 5-6 | The answer attempts to maintain a neutral tone, but there are noticeable lapses into subjective, slightly exaggerated, or emotionally tinged language in several places, which moderately detracts from the overall objectivity. |
| score 1-4 | The tone is frequently and significantly biased, subjective, sensationalist, or alarmist. Emotionally loaded language is common and severely compromises the objectivity and neutrality of the answer. |
| score 0 | The tone is entirely inappropriate, propagandistic, or completely fails to adopt a factual and objective stance. The answer is unrateable on its epistemological merits due to its tone. |

**Accuracy**

| | |
|---|---|
| description | Assesses the factual correctness of the information provided, its contextual relevance, and its basis in current scientific understanding. |
| score 9-10 | All information presented is entirely accurate according to current, mainstream scientific understanding. All scientific findings are presented in their proper context, without contradictions or reliance on anecdotal evidence. The information reflects the highest scientific standards. |
| score 7-8 | The vast majority of the information is accurate and well-contextualized. There may be a minor factual error or a slightly out-of-context statement that does not undermine the core argument or the overall correctness of the answer. |
| score 5-6 | The answer contains some accurate information but also includes several noticeable factual errors, misinterpretations, or statements presented out of their proper scientific context. The answer may contain minor self-contradictions. |
| score 1-4 | The answer contains significant, numerous, or fundamental factual errors. Information is frequently presented out of context, is self-contradictory, or relies on discredited science or anecdotal evidence. The core claims are scientifically unsound. |
| score 0 | The answer is completely factually incorrect, based on pseudoscience or misinformation, or provides no verifiable information. |

**Specificity**

| | |
|---|---|
| description | Evaluates how strictly the answer adheres to the scope of the question, including any explicit constraints (e.g., temporal or geographical). |
| score 9-10 | The answer addresses only and exclusively the explicit question asked. It does not introduce related but irrelevant topics. Any elaboration serves only to deepen the detail on the specific topic. All specified contexts (e.g., timeframe, location) are strictly respected. |
| score 7-8 | The answer is highly focused on the question but may include a minor, brief deviation into a tangentially related topic that does not significantly detract from the main focus. It largely respects all specified contexts. |
| score 5-6 | The answer addresses the core question but also includes a noticeable amount of irrelevant information or broadens the scope beyond what was asked. It may partially neglect or misinterpret a specified context. |
| score 1-4 | The answer significantly deviates from the question, focusing largely on related but irrelevant topics. The core question is only superficially addressed or misunderstood. Specified contexts are largely ignored. |
| score 0 | The answer completely fails to address the question asked and is entirely off-topic. |

Table 7: Shared (non question specific) rubrics used for evaluation (Part 2).

**Completeness**

| | |
|---|---|
| description | Assesses whether the answer thoroughly addresses all parts of the question, providing sufficient detail and acknowledging all relevant perspectives. |
| score 9-10 | The answer comprehensively addresses all parts of the question, leaving no significant aspect unexamined. It provides a thorough level of detail, including relevant and well-chosen data and statistics. It accurately represents the established scientific consensus and acknowledges different significant scientific perspectives where applicable. |
| score 7-8 | The answer addresses all major parts of the question but may omit a minor aspect or provide slightly less detail than would be ideal in one area. The overall picture is still robust and well-supported. |
| score 5-6 | The answer addresses the main parts of the question but omits some important aspects, fails to provide sufficient detail, or lacks nuance. |
| score 1-4 | The answer is notably incomplete, omitting major parts of the question or providing only superficial, cursory information. Key details, data, or established scientific perspectives are missing. |
| score 0 | The answer is a fragment or completely fails to provide a substantive response to any part of the question. |

**Sources**

| | |
|---|---|
| description | Assesses the quality, appropriateness, and proper citation of the sources used to support the answer. |
| score 9-10 | Every key point, statement, statistic, image or piece of data is meticulously supported by a traceable citation linked to the reference section. Sources are overwhelmingly first-hand, peer-reviewed academic literature or reports from major scientific bodies (e.g., IPCC, Nature, WHO). All sources are authoritative, up-to-date, and include a valid, working URL. |
| score 7-8 | Most key points are properly cited with high-quality, authoritative sources. There may be a few minor statements lacking a direct citation, or a small number of citations may be to high-quality secondary sources (e.g., reputable scientific journalism or a good Wikipedia article) instead of primary literature. All URLs are valid. |
| score 5-6 | Citations are present but inconsistent; some key claims lack support. The answer relies significantly on secondary sources, news articles, or institutional websites rather than primary academic literature. Some sources may be slightly outdated. |
| score 1-4 | Citations are largely missing, incorrect, or link to unreliable sources (e.g., blogs, opinion pieces, non-scientific websites). There is little to no use of peer-reviewed literature. Many key claims are unsupported. |
| score 0 | The answer provides no sources or citations whatsoever, or the sources provided are entirely inappropriate and untrustworthy. |

F.1.2 ANSWERS

---

**Model answer (Preferred by Scientists)**

The optimal data and measurement tools for climate risk modeling and disclosure depend on the specific type of risk being analyzed-physical or transition-and the intended application, such as internal strategy or external reporting. A combination of datasets and analytical tools is typically required rather than a single "best" solution [2].

**Data for Climate Risk Modeling**

Effective modeling requires both climate-related and entity-specific data.

**For Physical Risk Assessment:**
- **Climate Hazard Data:** Projections of chronic changes (e.g., sea-level rise, mean temperature shifts) and acute events (e.g., hurricanes, floods, wildfires) are fundamental. This data is derived from Global Climate Models (GCMs) and, for higher geographic resolution, dynamically or statistically downscaled Regional Climate Models (RCMs). These models use standardized scenarios, such as the IPCC's Shared Socioeconomic Pathways (SSPs), to project outcomes under different emissions futures [1]. Key data providers include the Copernicus Climate Change Service (C3S) and NASA.
- **Exposure and Vulnerability Data:** High-quality analysis requires granular, asset-level data, including precise geolocations of facilities, infrastructure, and key supply chain points [2]. This is combined with vulnerability data, which describes an asset's susceptibility to a given hazard (e.g., building elevation, flood defense standards) [3].

**For Transition Risk Assessment:**
- **Scenario Data:** This includes projections on policy, technology, and market shifts. Authoritative sources provide scenarios detailing potential pathways for carbon prices, energy mixes, and technology adoption rates. The Network for Greening the Financial System (NGFS) and the International Energy Agency (IEA) are key providers of such scenarios [3, 4].
- **Entity-Specific Data:** For both modeling and disclosure, companies must use internal data on their greenhouse gas (GHG) emissions (Scope 1, 2, and 3), energy consumption, revenue dependencies, and strategic plans for decarbonization. This is a core requirement of disclosure standards like IFRS S2 [5].

**Measurement Tools**
- **Scenario Analysis:** This is the overarching framework recommended by the Task Force on Climate-related Financial Disclosures (TCFD) for assessing and reporting on climate risks and opportunities. It involves using a set of plausible future scenarios to test the resilience of a business strategy [2].
- **Climate and Economic Models:** Integrated Assessment Models (IAMs) are a key tool for generating transition scenarios, as they link economic activity, energy systems, and the climate [6]. For physical risk, catastrophe (CAT) models, traditionally used in the insurance industry, are adapted to quantify the financial impacts of extreme weather events under climate change [7].
- **Financial Impact Models:** These tools translate the outputs from climate and economic models into financial metrics. This can involve adjusting standard financial models, such as Discounted Cash Flow (DCF) or Value-at-Risk (VaR) models, to incorporate climate-related variables [3].

*The image you are requesting does not exist or is no longer available. [BROKEN LINK]*

**Figure:** The analytical chain for climate risk assessment, showing the progression from scenarios to financial risk impacts. Source: Adapted from NGFS [3].

For disclosure, transparency is paramount. Organizations should report the specific models, scenarios, and key assumptions used in their analysis. It is also critical to acknowledge the inherent uncertainties in long-term climate and economic projections [1, 2].

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

**Hybrid answer (Human + Gemini 2.5 Flash) (Preferred by Autorater)**

Climate risk modeling and disclosure are essential for organizations to understand and report the potential financial impacts of climate change [1]. This involves assessing both physical risks, such as extreme weather events and sea-level rise, and transition risks, including policy changes, technological advancements, and market shifts [1]. Physical risk is fundamentally a combination of climate hazard, exposure, and vulnerability [2].

Effective climate risk modeling necessitates the use of diverse and high-quality data. For physical risks, crucial data types include climate science data (projections from climate models), geographic information, historical loss data [1, 2], observations, reanalyses, and climate indices like the Standardized Precipitation Index (SPI) or Standardized Precipitation Evapotranspiration Index (SPEI) [2]. High-resolution and even hyper-local data are increasingly important for assessing localized impacts [6, 2]. For transition risks, data encompasses information on policies, technology trends, market dynamics, and company-specific operational data such as energy use and emissions [1]. The quality, granularity, and forward-looking nature of data are critical for robust modeling [1, 3, 2]. Relevant data sources include outputs from climate models, remote sensing, ground observations [2], data from organizations like CDP [7], and company financial filings (e.g., 10-K reports) [5].

Measurement tools employed in climate risk modeling span a wide range of approaches. These include statistical models, scenario analysis, and stress testing [1, 3]. More complex tools involve integrated climate and economic models [1]. To translate large-scale climate data to local impacts, downscaling techniques, both dynamical and statistical, are frequently used [2, 6]. Textual analysis of corporate disclosures is also utilized to assess reported risks [5]. A conceptual framework can help outline the necessary data and appropriate risk measurement approaches (Figure 1) [4, 3].

*Image: Climate Risk Assessment Framework for Banks and Supervisors* [ImageLink]

**Figure:** A conceptual framework for climate risk assessment outlines key steps, including identifying data needs and applying risk measurement approaches [4].

Significant challenges exist in climate risk modeling and disclosure. These include issues with data availability, quality, consistency, and scalability [1, 3, 2]. Modeling inherently

involves uncertainties, particularly over longer time horizons [1, 2, 3]. Furthermore, there are challenges related to the transparency and comparability of the tools and the resulting disclosures [3]. Many tools lack public transparency regarding their setup and underlying assumptions, and there is a need for improved communication of output uncertainties and limitations [3]. The lack of peer-reviewed approaches for some tools also poses a challenge [3].

Ultimately, there is no single "best" type of data or measurement tool. Effective climate risk modeling and disclosure require integrating diverse data sources and applying appropriate tools tailored to the specific risks being assessed, the sector, and relevant disclosure standards [1, 3]. Forward-looking assessments and scenario analysis remain key components of a robust approach [1, 2, 3].

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

### F.1.3 VERDICTS AND AUTORATER RATIONALE

**Scientists Verdict:**  Model answer wins over Hybrid.
Dimensions favoring model answer:

- presentation_style
- presentation_clarity

**Autorater Verdict:**  Hybrid wins over model answer.
Dimensions favoring model answer:

- knowledge_specificity.

Dimensions favoring Hybrid:

- knowledge_uncertainty
- knowledge_accuracy
- knowledge_completeness
- citations_source_quality

**Autorater Explanation (Excerpt)**

"Answer 2 is preferred because it offers a more comprehensive and nuanced understanding of climate risk modeling and disclosure. While Answer 1 (model answer) provides excellent specificity regarding certain data types and models, **Answer 2 significantly surpasses it in its detailed acknowledgment and discussion of the inherent challenges, uncertainties, and limitations within this**

**complex field.** This crucial aspect is vital for a complete and realistic assessment of climate risk. Additionally, Answer 2 demonstrates better source quality... Answer 1's reliance on invalid references for key frameworks like NGFS and IFRS S2 weakens the verifiability of important details..."

Note that above Answer 1 refers to the model answer and Answer 2 refers to the Hybrid answer.

**Key Rubric Criterion: Nuance and Acknowledgment of Challenges (Excerpt)**

- **Gemini Pro Rating (5.0/10):** "Answer 1 briefly acknowledges 'inherent uncertainties...' in its concluding paragraph, but does not elaborate further."

- **Hybrid Rating (9.5/10):** "Answer 2 dedicates a significant section to 'Significant challenges,' discussing data availability, quality, consistency, scalability, inherent uncertainties (long time horizons), transparency, comparability, and the lack of peer-reviewed approaches. This provides a sophisticated and detailed discussion."

**Supplemental Materials Check (Excerpt)**

- The reference URL for IFRS S2 (*Used by model answer, Ref [5]*) is NOT valid.

- The reference URL for NGFS (*Used by model answer, Ref [3]*) is NOT valid.

- Answer 1 (*model answer*) contains 7 references. 2 of the 7 references NOT valid.

- Answer 2 (*Hybrid*) contains 7 references. 1 of the 7 references NOT valid (Ref [2], excluding local PDF links).

In the items above, the information regarding model identities in *italics* has been added for readability. The source for any answer is never available to the autorater.

