# OpenReview forum: "CLINB: A Climate Intelligence Benchmark for Foundational Models"
_ICLR.cc/2026/Conference — Submitted to ICLR 2026_

### Official Review · Reviewer_PS8B · 2025-10-31

**Soundness:** 3
**Presentation:** 2
**Contribution:** 2
**Rating:** 4
**Confidence:** 3

**Summary:**

This paper introduces CLINB, a benchmark designed to evaluate LLMs on open-ended, multimodal question-answering tasks in the context of climate science. The benchmark is constructed through a multi-phase, human-in-the-loop process involving climate scientists, experts, and advocates. It combines real user questions, curated rubrics, and model-based evaluation to assess models on both knowledge synthesis and grounding.

**Strengths:**

1. The authors convincingly argue that existing benchmarks fail to capture real-world, open-ended scientific reasoning tasks. CLINB fills a clear gap for evaluating AI in climate research and science communication.


2. The multi-stage, expert-in-the-loop design, especially the inclusion of domain scientists and advocates, adds transparency and credibility to the dataset and evaluation process.

3. The paper evaluates multiple frontier models with both human and LLM judges, showing interesting connections between human and automated scoring.

**Weaknesses:**

1. The models used in the experiments are not vast enough. For example, there lack of sufficient treatments of open-source LLMs.


2. The data pipelines involve too many experts, making the methodology hard to scale up and reproduce.



3. It is unclear how to use the benchmark data to improve the performance of LLMs on climate topics.

4. The metric used is mainly the ELO score, some more absolute metrics like accuracies should be considered.

**Questions:**

See Weakness.

---

### Official Review · Reviewer_Qgoh · 2025-11-01

**Soundness:** 2
**Presentation:** 2
**Contribution:** 3
**Rating:** 4
**Confidence:** 3

**Summary:**

The paper introduces CLINB, a new benchmark to evaluate how well Large Language Models handle complex, specialized knowledge, using climate change as the test domain. It uses real user questions and expert-curated rubrics to assess models on open-ended, multimodal, and grounded question answering. The main finding is a critical dichotomy: frontier models demonstrate remarkable, PhD-level knowledge synthesis but suffer from major failures in grounding, with high rates of hallucinated references and images. The study also validates a model-based evaluation process and finds that top autonomous models can outperform "hybrid" answers created by domain experts who were assisted by weaker models.

**Strengths:**

I think the primary strength of this paper is its rigorous methodology. It moves beyond simple, closed-form benchmarks to tackle the much harder problem of evaluating open-ended, specialized scientific communication. The three-phase data creation process, involving multiple tiers of human experts and scientists, is thorough. It directly measures a critical and often-overlooked failure point in LLMs: the gap between fluent synthesis and verifiable grounding. The validation of its "autorater" against top-tier scientist judgments is also a strong contribution, and it is transparent about its limitations. This is the finding I like the most that models are great at sounding smart (synthesis) but terrible at proving they're right (citations). In science, medicine, or law, an answer without a valid source is useless (also particularly true in tasks like fact-checking where evidence is essential).

Therefore although seems niche, I am interested to see such work.

**Weaknesses:**

However, I think the main weakness is the scalability of its own methodology. The creation of the expert-driven, question-specific rubrics is resource-intensive, and the paper notes that these rubrics may not generalize well to future models. This raises questions about how CLINB can be sustainably updated. This is the major concern that I have, especially for a dataset paper.

the paper's conclusion that "familiarity bias" caused the disagreement between rater groups (Experts vs. Scientists) is a strong interpretation. the disagreement could also stem from different-but-valid weighting of answer qualities, which is not fully disproven.

Given these weaknesses, questions (below) and ethic considerations (below), I am leaning towards a weak reject. As the authors themselves noted in the paper this is not an average crowd-sourcing dataset. However I strongly encourage the authors to continue working on the automatic data construction, even just semi-automatic would be very helpful!

**Questions:**

Given the high cost of creating the expert rubrics, how do you envision keeping the benchmark relevant as new models are released? Is there a semi-automated way to update the rubrics?

Your finding that motivated 'Advocates' produced better answers than 'Experts' is counter-intuitive. Could this be less about expert motivation and more about the AI-assisted 'Editor' tool being poorly suited to an expert's existing workflow?

How confident are you in the 'familiarity bias' explanation for the rater disagreement? Is it not possible the 'Experts' were justifiably prioritizing visual content and presentation, which the 'Scientists' later de-prioritized?

**Details Of Ethics Concerns:**

The paper's entire methodology hinges on the intensive labor of 57 highly-skilled human raters (40 "Experts" and 17 "Advocates"). This is not a simple crowd-sourcing task as it's a "knowledge intensive" process that required recruiting active academics (PhDs and postdocs) and dedicated members of a specialized NGO. The paper fails to address any aspect of compensation for this labor. There is no mention of whether the Experts or Advocates were paid for their time, and if so, at what rate (the total costs is also unknown). Given that the paper acknowledges this expertise is "a scarce resource", the ethical responsibility to provide fair compensation for such high-level work is critical. Omitting this information is a significant gap in responsible research practice. The reliance on what may be uncompensated or under-compensated expert labor to build a benchmark might be an ethical issue that should have been addressed. An ethics review might be needed in this case.

---

### Official Review · Reviewer_tWt5 · 2025-11-01

**Soundness:** 2
**Presentation:** 2
**Contribution:** 2
**Rating:** 2
**Confidence:** 4

**Summary:**

The paper presents a new benchmark namely CLINB to evaluate LLMs on complex climate science questions. It combines real user queries, expert-validated rubrics, and human–AI collaboration. Results show that frontier models like GPT-5 and Claude Opus 4.1 achieve PhD-level knowledge synthesis but still struggle with evidence grounding, often hallucinating citations and images. The work highlights the need to bridge this gap to build trustworthy, verifiable scientific AI.

**Strengths:**

* This paper introduces the first expert-validated, open-ended, multimodal benchmark focused on climate intelligence, moving beyond traditional multiple-choice or trivia-style tests.

* Unlike traditional multiple-choice or trivia-style evaluations, CLINB focuses on realistic, evidence-based scientific reasoning. The benchmark is built through collaboration among scientists, domain experts, and informed non-experts, which ensures both diversity and data quality. I especially appreciate the emphasis on verifiable references and images, bringing the evaluation closer to real-world scientific communication.

* The proposed rubric-guided, model-based evaluation pipeline is also well designed.

* Overall, the findings are insightful, showing that frontier models like GPT-5 and Claude Opus 4.1 demonstrate PhD-level synthesis but still struggle with grounding, while this conclusion is already extensively revealed by recent studies, such as [1].

[1] GDPval: Evaluating AI Model Performance on Real-World Economically Valuable Tasks.

**Weaknesses:**

1. While the author argue its the first work on climate evalution work, there are still some relevant datasets already exists. [1] [2] The author didn't discuss these related works and compare the difference.

2. This paper claims that the model has reached a PhD-level capability, yet it still exhibits significant hallucinations in grounding. The two points are contradictory, which makes the argument unconvincing.

3. There is not any methodology novelty. Even though the zero-shot prompting is not extensive, and not including

4. Although the paper highlights grounding failures (with a hallucination rate as high as 25%), it does not delve into which types of cited sources are most prone to hallucination or the possible causes of these hallucinations. The experimental analysis is not sufficiently in-depth.

5. CLINB is designed with an emphasis on “trustworthy evaluation,” but the experimental phase completely excludes open-source models. As a result, while its evaluation reflects the capabilities of leading commercial systems, it lacks a basis for verification and comparison within the scientific community.

Overall, the contributed dataset is valuable, but the evaluation and analysis is still weak and not ready for publication.

[1] Climate-Eval: A Comprehensive Benchmark for NLP Tasks Related to Climate Change.

[2] Assessing Large Language Models on Climate Information.

**Questions:**

* Do the authors plan to extend CLINB to include evaluations of open-source models? If not, how can this benchmark achieve true open science and comparability?

* How do the authors verify the fairness and stability of the automatic evaluation system? Have they tested the impact of using different judge models or different rubrics on the results?

---

### Official Review · Reviewer_CRXE · 2025-11-01

**Soundness:** 3
**Presentation:** 3
**Contribution:** 3
**Rating:** 6
**Confidence:** 2

**Summary:**

The authors develop the CLINB benchmark to assess models on open-ended, grounded, multimodal climate-related question-answering tasks. One important advantage of their benchmark is that it involves real users' questions and evaluation rubrics developed by domain experts rather than arbitrary scoring strategies. They conclude that frontier models can surpass the hybrid solutions curated by domain experts assisted by weaker models. However, there are issues with hallucination rates for references and images that need to be addressed for reliable deployment of AI in scientific workflows.

**Strengths:**

The benchmark includes complex questions that demand research, assessment of evidence, and careful synthesis, and it expects long-form answers that may include visuals and must be grounded with references and citations. The evaluation is strengthened by feedback from domain experts at each stage, with detailed grading rubrics that are created and verified, and scientists who contribute to data creation and check the scientific validity of the results.

**Weaknesses:**

(1) The paper evaluates references and visuals by checking whether links exist, not whether the cited content is correct, relevant, or high quality. This can make an answer appear grounded even when the source does not support the claim or the visual is mismatched.

(2) The hybrid answer generation uses a weaker base model, and the rationale for this choice is not fully explained. The hybrids that use stronger base models, or at least a clear standalone baseline for the weaker model in the same setting, are not provided.

(3) Minor issue: In Figures 3 and 9, the shared legend for link status appears attached to the other subplot, which is unintuitive.

(4) It is mentioned that the autorater's image scores for GPT-5 look optimistic even when the model does not provide image links. This suggests that strong text answers may be raising the image score. If the image score is meant to reflect only visual evidence quality, it should be isolated from other aspects of the response and computed only when verifiable image references are present.

**Questions:**

(1) The informal analysis of strengths and weaknesses, including the category labels in Table 2, does not describe its procedure or reliability. Could you please expand on this part?

(2) Have you considered verifying the content of cited sources and visuals, not only whether links resolve? For example, you could use a rubric-guided LLM as a judge, expert validation, retrieval consistency tests, or scoring of citation appropriateness and visual correctness on a sampled set.

(3) Could you clarify why hybrid answers are presented only with a weaker base model? Would you consider reporting hybrids built on stronger models, and also the weaker model on its own?

---

### Author Response · Authors · 2025-12-03
**Response to the reviewers**

We thank all reviewers for their time and constructive technical comments which we sincerely appreciated. We acknowledge the valid limitations pointed out in the reviews which we did not have time to address during the rebuttal period. They will help us strengthen the next iteration of this work.

* Compensation: All participating raters were fairly compensated.

* The main reason for using Gemini 2.5 Flash (a "weaker model") to generate hybrid answers was latency. Due to long multimodal inputs and outputs, the response time was too slow for a responsive UI.

* Grounding vs. Quality: Regarding the observation that frontier LLMs produce PhD-level content but suboptimal grounding: while this seems somewhat contradictory, it is purely descriptive. This is what we observed; indeed, the PhD-level assessment is supported by the scientists.

* Novelty: With respect to the data generation process, we believe the multi-stage pipeline makes genuinely novel contributions, such as the graph-based, data-driven rubric generation. We also emphasize that we conduct expert human validation at each step. We have added a comparison table against ClimaQA, AtmosSci‑Bench, Bulian et al., and ClimateEval, to highlight the differences.

* With regard to the familiarity bias hypothesis concerning the disagreement between raters and autorater/scientists: it is possible that the experts rate the presence of images more highly, because they spent significant amounts of time curating answers that contain images.

* With respect to the fact that Advocates can produce better answers than Experts because they engage more: the review suggests that the tool is poorly suited to an expert workflow and that may be the reason why Advocates do a better job. However, we have no evidence that the tool was less suited to experts, or that it is more suited for non-experts. Also, the scientists (the most experts of all) were able to use it productively.

* Regarding absolute scores, i.e. pointwise ratings: We agree they are crucial alongside the relative rankings. Our choice to prioritize Elo scores (based on pairwise comparisons) was mainly driven by reliability. When dealing with these complex, long-form answers, we found that comparative judgments are significantly more consistent than asking the autorater to assign a standalone score. The pairwise framework is also more suited to evaluating new, improved models that may be out-of-distribution with the existing rating scale. That being said, we're planning to add a pointwise evaluation to further ground the analysis of specific models' strengths and weaknesses.

* With respect to the observations about scalability, hybrid answers with stronger models, evaluating OS systems and frontier models that came out after the submission deadline, and benchmark utility. We will include these in the next revision of the paper.

Thanks again!

---

### Meta-Review · Area_Chair_A9X5 · 2025-12-28

**Summary:**

The reviewers raised valid concerns regarding the exclusion of open-source models (Reviewers tWt5, PS8B), the scalability and reproducibility of the expert-driven data pipeline (Reviewers Qgoh, PS8B), and the validity of evaluation metrics that rely on link checking rather than content verification (Reviewer CRXE). The authors provided a single, generalized response rather than detailed replies to specific questions. Crucially, the authors explicitly admitted that they "did not have time to address" valid limitations during the rebuttal and stated that the feedback would help the "next iteration" of the work. Consequently, major concerns remain unaddressed in the current submission.

**Reviewer Concerns:**

**Addressed Concerns:**
* **Reviewer Qgoh's** ethical concern regarding annotator compensation was addressed (authors confirmed raters were fairly compensated).
* **Reviewer CRXE's** query regarding the rationale for using a weaker base model for hybrid answers was explained (latency constraints).

**Outstanding Concerns:**
* **Reviewers tWt5 and PS8B's** request for open-source model evaluations was not addressed; the authors only promised to include this in the "next revision."
* **Reviewers Qgoh and PS8B's** concerns regarding the scalability of the rubric creation process were not resolved; the authors did not provide any concrete solution or alternative method.
* **Reviewer CRXE's** specific questions regarding content verification (vs. simple link resolution) and the reliability of the informal analysis were not addressed.
* **Reviewer PS8B's** suggestion to include absolute accuracy metrics alongside ELO scores was deferred to future work.

**Reviewer Scores:**

* **Reviewer CRXE:** 6 -> 6. While the reviewer was initially positive, the authors failed to answer specific questions regarding content verification and informal analysis reliability. The score is unlikely to improve.
* **Reviewer tWt5:** 2 -> 2. The reviewer gave a score of 2 due to concerns such as the lack of open-source models and deep analysis. The authors explicitly stated they would address these in the "next revision," confirming the current version remains insufficient.
* **Reviewer Qgoh:** 4 -> 4. Although the ethical concern regarding compensation was resolved, the reviewer's "major concern" regarding the scalability of the methodology was not addressed. The general response is unlikely to alleviate the concern about sustainable updates to the benchmark.
* **Reviewer PS8B:** 4 -> 4. The reviewer's primary concerns regarding the limited model set (lack of open-source models) and scalability were deferred by the authors to a future version.

---

### Decision · Program_Chairs · 2026-01-26

Reject